# Dynamic signature of activity-stability tradeoff in lactamase evolution

Ernesto Arcia[1,5], Dimitra Keramisanou[1,5], Lian M. C. Jacobs[2], McKenna Parker[1], Julián M. Delgado [3], Vasantha Kumar MV[1], Sameer Varma [3], Rinat Abzalimov[4], Yu Chen [2] ✉ & Ioannis Gelis [1] ✉

Our ability to understand protein evolution hinges on understanding how evolutionary landscapes are shaped at the fundamental protein level. Using TEM-1 β-lactamase we show that molecular traits related to the statistical ensemble nature of protein structure contribute to broader substrate specificity, active site-scaffold communication, and the selection of stabilizing substitutions. During the evolution of cefotaxime resistance, the initial mutation reorganizes the active site, introducing a new function conformation. Secondary substitutions improve catalytic efficiency by redistributing the ensemble and restoring a significant population of the original conformation, rather than by stabilizing the new conformation. Stability defects associated with initial mutations are not evenly disseminated but are clustered at specific distal scaffold elements. The capacity of mutants to independently modulate the populations of individual active site walls and scaffold regions through narrow residue networks, produces conformational epistasis and a combinatorial set of cefotaximase states, which enables local compensation of scaffold defects.

Adaptive protein evolution occurs through successive mutation steps that define an evolutionary landscape towards increased fitness. Understanding the sequence-structure-function relationship at the protein level is critical for predicting evolution, designing distinct functions in existing enzymes, and engineer effective biologics. During the evolution of TEM-1 from a bona fide penicillinase to an extended-spectrum β-lactamase (ESBL) with broad activity against third generation β-lactam cephalosporins, such as cefotaxime and ceftazidime, a set of highly beneficial mutations is initially fixed at loops or termini of secondary structure elements flanking the active site[1]. These mutations typically have a strong impact on the enzyme's catalytic properties. This is followed by secondary mutations distributed throughout the protein fold, at either distal or interacting sites relative to the initial mutation[1]. Secondary mutations have weak to no effect on catalysis but alter the enzyme's biophysical properties, reversing detrimental pleiotropic effects caused by the initial mutations and optimizing fitness. This temporal and spatial pattern of evolutionary steps[2–4], forms the basis of activity-stability tradeoff, a pervasive mechanistic constraint in protein evolution[5,6] that reflects the balance between active site functional innovation and scaffold robustness[7].

Initial mutations provide functional innovation by acting upon the underlying enzymatic promiscuity, increasing fitness by affecting a functional property, such as the $k_{cat}/K_M$. They do so by shifting the conformational manifold of the active site to sparsely populated conformers or by allowing for de novo sampling of conformers that confer new specificities and functions[8–15]. However, random mutations are destabilizing by a $\Delta\Delta G$ factor of 0.5−5.0 kcal/mol, which is of the same order as $\Delta G_{initial}$, while "new function" mutations at the active site result in even higher $\Delta\Delta G$ values[5,16–24]. Evidently, for protein sequences with low robustness or when successive evolutionary steps produce

[1]Department of Chemistry, University of South Florida, Tampa, FL, USA. [2]Department of Molecular Medicine, Morsani College of Medicine, University of South Florida, Tampa, FL, USA. [3]Department of Molecular Biosciences, University of South Florida, Tampa, FL, USA. [4]Structural Biology Initiative, CUNY Advanced Science Research Center, New York, NY, USA. [5]These authors contributed equally: Ernesto Arcia, Dimitra Keramisanou. ✉e-mail: ychen1@usf.edu; igelis@usf.edu

large positive $\Delta\Delta G$, or even when non-native intermediate states are populated, secondary compensatory[25–29] or ancestor mutations[16,25,29–33] are fixed to restore scaffold stability and avoid exhausting the $\Delta\Delta G$ robustness window. These secondary mutations typically act in an epistatic manner, being neutral in the initial background but beneficial in the mutant background[25,34].

Although the tradeoff model forms the biophysical basis for the two opposing selective pressures, it does not provide a complete molecular view of how the mechanism-based observables $\Delta\Delta G$ and $k_{cat}/K_M$ render specific evolutionary trajectories accessible. For TEM-1, for example, $\Delta G_{N-U}$ has no explanatory power in models using minimal inhibitory concentration (MIC) as a fitness proxy, in contrast to $k_{cat}/K_M$, which drives the evolution of cefotaxime resistance[35]. Additionally, thermal unfolding may not capture all fundamental scaffold properties that act as selective pressure for secondary mutations, including non-native states[6,36] or kinetic stability effects[37,38]. Importantly, neither thermal nor chemical unfolding can resolve the energetic coupling between local and remote sites, which imposes spatial contingencies of energy propagation and explains how secondary substitutions are fixed at a particular site in a given genetic background[39,40]. Finally, structural information alone, without insights into conformational dynamics, fails to provide the complete set of molecular traits behind $k_{cat}/K_M$-driven selection and the acquisition of initial beneficial mutations[39–41].

Here, we investigate a set of TEM-1 evolutionary intermediates involved in acquiring the ESBL phenotype. TEM-1 is an ideal model system to characterize enzyme evolution since there is ample biochemical and biophysical data available providing a detailed map of its adaptive landscape (Supplementary Table S1). Importantly, it has a clear functional context, free of metabolic cross-effects, which allows for a direct link between genetic changes, phenotypic outcomes and molecular properties, including $\Delta G_{N-U}$ or $k_{cat}/K_M$. We find that along an evolutionary landscape the active site walls of TEM-1 participate in a dynamic two-state population shift between penicillinase and cefotaximase optimized states. Individual mutations control the population of each active site wall conformation, giving rise to conformational epistasis and a combinatorial set of states. The new function conformation induced by the prominent G238S substitution is not progressively stabilized relative to the original penicillinase conformation, but secondary substitutions optimize the equilibrium populations of the two conformations. The compromised stability associated with G238S is not evenly disseminated but instead is clustered at specific distal scaffold sites. Secondary substitutions fixed within those regions locally correct these defects through narrow networks. Overall, we show that molecular proxies relevant to the statistical ensemble nature of protein structure explain how adaptive substitutions are selected within the activity-stability tradeoff constraint.

## Results

### G238S has a global effect on the conformational properties of the β-lactamase fold

The active site walls of class A β-lactamases are formed by residues located at the termini of secondary structure elements, including helix $\alpha_2$ (69-73), strand $\beta_3$ (230-237), and helix $\alpha_{12}$ (272, 275, 276), as well as loops 103-107 (105-loop), 130-132 (SDN-loop), 164-178 (Ω-loop), 213-220 (216-loop), 238-243 (238-loop) and 267-272 (270-loop), according to the Ambler numbering scheme[42] (Supplementary Fig. 1). Mutation of any of the core residues S70, K73, S130/N132, E166/N170 has a detrimental effect on function, hence adaptive benefits are realized by initial mutations in other positions within these segments. Despite the vast sequence space available, the most prominent pathways for the evolution of TEM-1 ESBL phenotype include the G238S substitution[1,43] (Fig. 1a). G238S (TEM-19) causes a ~100-fold increase in the $k_{cat}/K_M$ for cefotaxime hydrolysis, while maintaining high levels of the native

penicillinase activity, albeit with a ~10-fold reduction in $k_{cat}/K_M$ for ampicillin[6,44].

To understand the molecular mechanism by which 238-loop substitutions modulate hydrolysis of bulky cephalosporins, we used NMR spectroscopy. The backbone amide signals in a $^1$H-$^{15}$N-HSQC spectrum are site-specific probes sensitive to local conformational properties, reporting on both structural and dynamic features. Comparison of the TEM-1 and G238S spectra (Fig. 1b) reveals widespread chemical shift perturbations (CSPs) and signal attenuation (Fig. 1c), suggesting that the mutation elicits conformational changes that reach well beyond the mutation site. The regions showing prominent changes (Fig. 1d) include residues flanking the substitution site at the 238-loop, as well as the entire Ω-loop sequence, which is directly implicated in catalysis by providing the catalytic E166 and N170. Most 238-loop residues remain unassigned due to severe exchange broadening, while three Ω-loop residues are tentatively assigned based on signal proximity, and one remained unassigned. This is consistent with the dynamic structural rearrangements observed in the G238S crystal structures[6,45–47], where the 238-loop samples two states: a low-populated TEM-1-like "closed" conformation, competent for catalysis, and an "open" conformation, with the conserved E240 ($^N$H) · N170 (O) hydrogen bond that pins the Ω- and 238-loops disrupted, resulting in an expanded active site (Fig. 1e). Significant perturbation is also observed for the catalytic S130 and S70, the latter of which covalently binds β-lactams in the acylation state and is tentatively assigned in G238S. Furthermore, signals from residues comprising other active site walls, including helix $\alpha_2$, the SDN-, 216- and 270-loops, show significant attenuation due to exchange broadening (Fig. 1c, d). Thus, in addition to the 238-loop dynamics revealed by crystallographic studies, that evidently affect even subtly the Ω-loop conformation, the spectral properties of G238S show that the acquired cefotaximase activity is associated with enhanced conformational dynamics throughout the active site occurring at the fast-to-intermediate NMR timescale. Notably, CSPs and line broadening effects are propagated beyond the active site. Changes are observed for residues located at strands $\beta_1$, and $\beta_3$-$\beta_5$, which make up the core of the α/β subdomain, the C-terminal end of helix $\alpha_1$, a large part of helix $\alpha_{12}$, the corresponding loops, and even residues 27 Å away from the mutation site (Fig. 1c,d), at positions with very low RMSD between TEM-1 and G238S structures (Fig. 1c).

Therefore, in addition to the well documented difference observed in the architecture of the G238-loop, solution-based techniques reveal that the G238S induced conformational changes are propagated to all active site walls and remote scaffold sites. These changes are not captured in the available crystal structures, either because structures were determined in the presence of designed or secondary stabilizing substitutions, or because they reflect subtle dynamic changes that are only captured by solution-based techniques.

### G238S enhances the microsecond-millisecond timescale dynamics of TEM-1

To date, the molecular features that convert G238S or other initial substitutions into extended spectrum β-lactamases are not fully understood, mainly because the structures of these evolutionary intermediates have not been determined in a wild-type background. Instead, the effect of G238S on the conformational properties of TEM-1 has been studied in clinical isolates with acquired secondary substitutions, including TEM-52 (E104K/M182T/G238S)[45] and TEM-72 (Q39K/M182T/G238S/E240K)[47], or in the presence of multiple laboratory-evolved stabilizing substitutions, which facilitate recombinant expression and crystallization of the enzyme (G238S-TEM-v.13)[29,46]. The consensus of these studies is that the ESBL phenotype is linked to an enlarged active site, caused by alternative loop conformations, consistent with the structures of other extended-spectrum enzymes like Toho-1[48]. However, no distinct loop conformation has

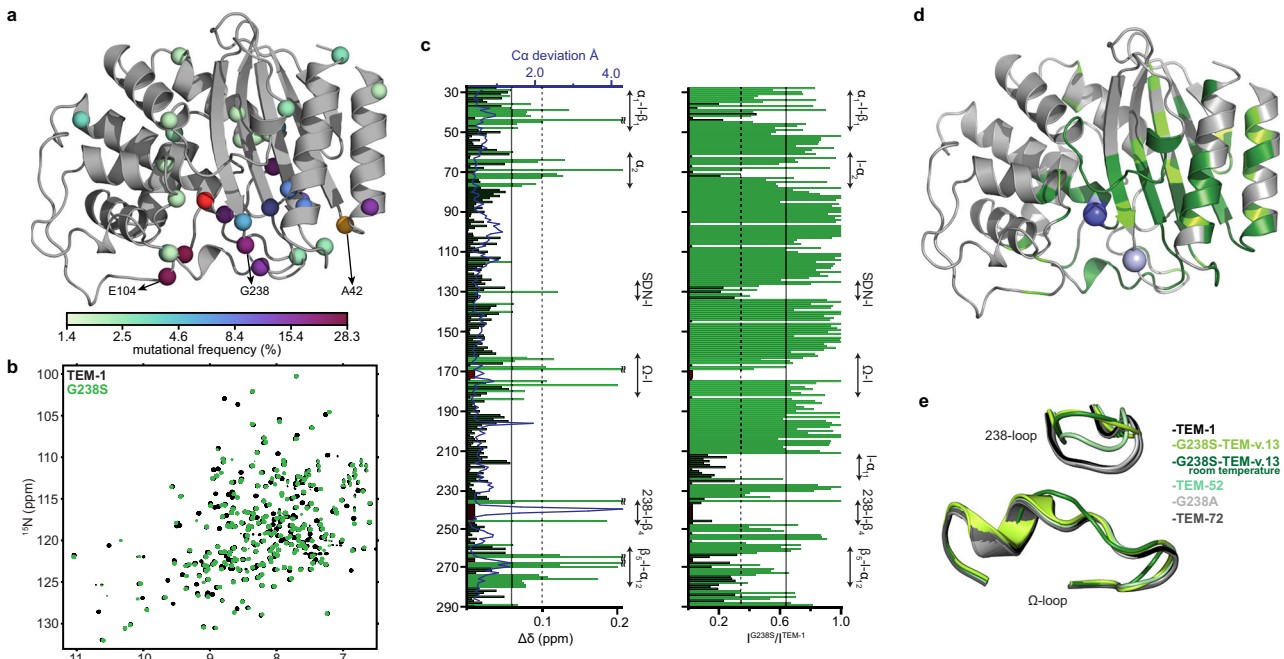

**Fig. 1 | The initial substitution G238S has a global effect on the conformational properties of TEM-1. a** Percent mutational frequency of each TEM-1 site observed across all clinical TEM-1 isolates shown in a green-purple color gradient. The number of mutations identified in each allele, the order by which multiple mutations are acquired within an isolate and the nature of the new amino acid are not considered. The clinically relevant sites E104 and G238, and the laboratory evolved A42 (orange) studied in here are indicated, together with the catalytic S70 (red). **b** Overlay of the $^1$H-$^{15}$N-HSQC spectra of TEM-1 (black) and G238S (green) acquired at 30 °C. **c** Left: combined $^1$H and $^{15}$N chemical shift perturbations ($\Delta\delta$) observed for G238S relative to TEM-1 as a function of residue number (bottom axis). The continuous and dotted black lines mark the mean and one standard deviation above the mean. The blue line (top axis) represents the deviation of $C_\alpha$ atoms between TEM-1 (PDB 4IBX) and G238S (PDB 4OPR) structures. Red boxes mark unassigned and tentatively assigned residues, and black dots mark proline residues. Right:

attenuation of signal upon introducing G238S to TEM-1, expressed as intensity ratio $I^{G238S}$ / $I^{TEM-1}$. The continuous and dotted black lines mark the mean and one standard deviation below the mean. **d** Mapping of the effect of G238S on the structure of TEM-1. Secondary structure elements are indicated in Supplementary Fig. 1. Chemical shift perturbations above the mean and one standard deviation above the mean are shown in green and dark green, respectively. Signals broadened beyond detection or attenuated one standard deviation below the mean are also shown in dark green. The site of the mutation and the position of S70 are shown as light and dark blue spheres, respectively. **e** Comparison between the 238- and $\Omega$-loop conformations acquired in the presence of G238 substitutions and TEM-1. Shown are the crystal structures of TEM-1 (1ZG4), TEM-52 (E104K/M182T/G238S, 1HTZ), G238S-TEM-v.13 (4OPR), room temperature G238S-TEM-v.13 (4OP8), TEM-72 (Q39K/M182T/G238S/E240K, 3P98) and G238A-TEM-1 (1JWV).

been identified (Fig. 1e), suggesting that broadened substrate specificity and acquired cefotaximase activity result from conformational heterogeneity and enhanced active site dynamics. Specifically, TEM-72 and the Ala-substituted enzyme, G238A, cause only marginal changes to the 238- or $\Omega$-loop (Fig. 1e). In TEM-52, however, G238S expands the active site by displacing the tip of the 238-loop, while in G238S-TEM-v.13, it causes the dynamic sampling of two conformations, resulting in an even larger displacement of the 238-loop and a more expanded active site (Fig. 1e).

To identify the dynamic traits that accompany acquisition of the ESBL phenotype, we investigated the underlying protein motions of G238S. To this end, we used NMR to measure backbone $^{15}$N $R_1$ and $R_2$ relaxation rates, as well as [$^1$H]-$^{15}$N heteronuclear NOEs at two magnetic field strengths (Supplementary Fig. 2a). Optimization of the rotational diffusion tensor using these observables yields an ellipsoid model with a global correlation time $\tau_m = 12.7$ ns and an anisotropy $D_{||}/D_\perp = 1.16$ (Supplementary Fig. 2b, c). These values are very close to the parameters determined previously for TEM-1 and its variants[49–51], suggesting that the global properties of the two proteins are similar. To describe the effective local motions and identify changes in protein flexibility induced by G238S at the ps-ns timescale, the relaxation data were analyzed using the model-free approach and interpreted based on the squared order parameter, $S^2$. The order parameter describes the amplitude of the internal motion of an N-H bond vector in a cone and ranges from $S^2 = 1$ to $S^2 = 0$ for fully restricted and unrestricted motions, respectively. Similar to other $\beta$-lactamases studied by NMR,

including TEM-1, PSE-4, Toho-1, CTX-M, and a number of their mutants[49–55], the $S^2$ values of G238S reveal that the enzyme retains the rigidity observed for TEM-1, with an average $S^2$ of $0.84 \pm 0.01$ (Fig. 2a and Supplementary Table S2) and only small $|\Delta S^2|$ values (<0.2) (Supplementary Fig. 2d). As expected, the only two positions showing large $|\Delta S^2|$ values (> 0.2) are not essential for catalysis and include K32 and Q278 at helices $\alpha_1$ and $\alpha_{12}$, respectively (Supplementary Fig. 2e). The sidechains of these residues form a hydrogen bond that connects the two terminal helices, while the neighboring N276 links helix $\alpha_{12}$ to both the active site and the mutation site through a hydrogen bond with R244, at the C-terminus of the 238-loop, highlighting how active site substitutions may induce ps-ns timescale motions at remote regions of the scaffold (Supplementary Fig. 2e).

For many residues, proper fitting of the relaxation data required an additional term, $R_{ex}$, to account for the contribution of slower timescale motions to $R_2$ ($\mu$s-ms), as evident by the local model-free model selected for each site (Supplementary Table S2). In order to characterize the exchange properties at this timescale we acquired a series of Carr-Purcell-Meiboom-Gill (CPMG) relaxation dispersion experiments, where in the presence of conformational exchange the effective relaxation rate, $R_{2,eff}$, changes as a function of repetition frequency ($\nu_{cpmg}$) of a series of $\pi$ pulses within a constant time period (Fig. 2b). The set of G238S residues undergoing $\mu$s-ms conformational exchange ($R_{2,eff(50\ Hz)}$ - $R_{2,eff(2000\ Hz)} \geq 4$ s$^{-1}$) cluster primarily at the $\alpha/\beta$ subdomain of the lactamase fold (Fig. 2c, d and Supplementary Table S2). When signals missing due to excessive broadening are

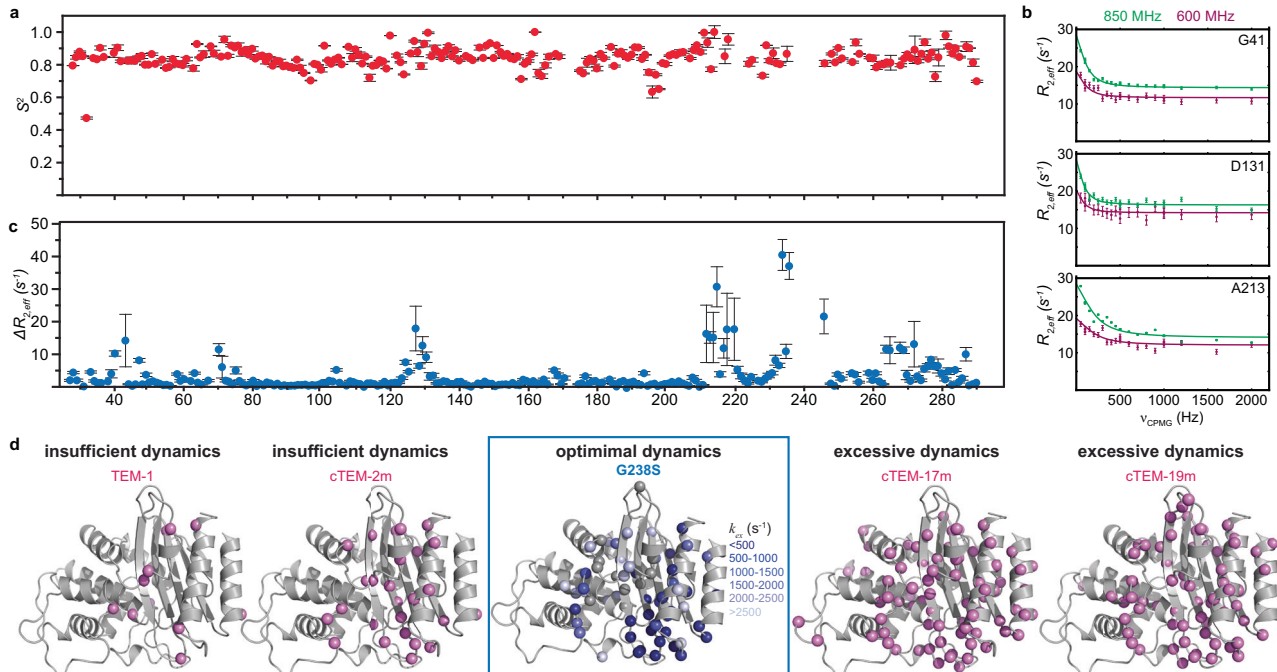

**Fig. 2 | G238S has a strong impact on the millisecond-timescale motions of TEM-1. a** Squared order parameters of G238S as a function of residue number reveals an overall rigid enzyme with no ps-ns timescale motions (average $S^2 \sim 0.85$). Errors are propagated from relaxation data measured in duplicate via Monte Carlo simulations. **b** Relaxation dispersion profiles for selected residues of G238S acquired at 600 MHz (purple) and 850 MHz (green) and 30 °C, showing $R_{2,eff}$ as a function of refocusing pulse train frequency, $\nu_{cpmg}$. Errors are propagated from relaxation data measured in duplicate via Monte Carlo simulations. **c** $\Delta R_{2,eff}$ from CPMG experiments of G238S, acquired at 850 MHZ, expressed as the $R_{2,eff}$ difference between $\nu_{CPMG}(50\,Hz) - \nu_{CPMG}(2000Hz)$. Errors in $\Delta R_{2,eff}$ were propagated according to standard error-propagation for a difference. **d** Mapping of the G238S sites showing μs-ms timescale motions are shown at the center of the panel, color-coded from dark to light blue according to their corresponding $k_{ex}$ (500–3000 s⁻¹) or gray if they show dispersion ($\Delta R_{2,eff} > 4\,Hz$) but are too broad for accurate fitting. Mapping of the corresponding sites of TEM-1 and engineered TEM-1 chimeras with enhanced or suppressed dynamics compared to G238S are shown for comparison.

considered in this timescale regime, a total of 64 G238S residues undergo slow dynamic conformational fluctuations (Fig. 2d). Except for the dispersion curves of M129, G269, A280, A284 and I287 that were best fitted with the general Carver-Richards equation (Supplementary Table S2), the dispersion data of all other residues were best described by the 2-state fast exchange Luz-Meiboom equation, where the apparent exchange rate, $k_{ex}$, is accurately determined, but the populations of the two states are not. Although the $k_{ex}$ for different G238S sites cover a wide range of values (Supplementary Table S2), when mapped on the structure, an asymmetric spatial distribution is evident (Fig. 2d). Active site residues undergo a slow exchange transition ( $< 500\,s^{-1}$) which is of the same order as the $k_{cat}$ of cefotaxime hydrolysis (15-85 s⁻¹), while scaffold residues experience significantly faster exchange transitions ( $> 2500\,s^{-1}$). Exchanging residues are distributed across all the active site walls (Fig. 2d). These include T71 and F72 at helix $\alpha_2$, which are adjacent to the catalytic S70, the complete 238-loop, as well as R275 from helix $\alpha_{12}$, which is one of the residues forming the oxyanion hole. In addition, residue Y105, which is involved in substrate recognition and binding, and contributes to active site expansion via alternate rotamer sampling, all residues from the SDN-loop, including the catalytic S130, the entire 216-loop, featuring the invariant V216, which stabilizes the position of large substrates[56], as well as all residues in the 270-loop, which is connected to the 238-loop via hydrogen bonding. Notably, only two residues in the Ω-loop, E168 ($R_{2,eff}$) and A172 (broadened), exhibit μs-ms conformational exchange, while the catalytically important residues E166 and N170 remain rigid. For E166, even the observed chemical shift change is minimal, suggesting that the mutation has little effect on the local conformation in this region of the Ω-loop. Compared to TEM-1 and a set of engineered TEM-1 enzymes studied previously by NMR[57,58] (Fig. 2d), the slow active site dynamics induced by G238S indicate that an evolutionary

optimized window of motions provides access to the new function, as neither excessive dynamics that include the Ω-loop wall and the catalytic E166 nor limited dynamics where the other active site walls are rigid, result in efficient cefotaximase activity.

In summary, G238S has a profound impact on the dynamic landscape of TEM-1. It induces extensive μs-ms dynamics at the active site walls and a hierarchic profile of motions at the scaffold, with enhanced ps-ns timescale dynamics at key sites linking secondary structure elements, causing widespread μs-ms dynamics.

## Dynamic population shift of TEM-1 conformers along cefotaximase evolution

To assess how secondary substitutions modulate the conformational landscape of TEM-1, we investigated the spectral properties of multiple mutants. These span the first three steps of the well-studied evolutionary pathway of TEM-1 to TEM-1* (TEM-1 → G238S → E104K → A42G → M182T), a model cefotaximase that increases cefotaxime MIC by a factor of ~32,000[59–61]. We omitted the last step (M182T), as it acts as a global suppressor[62,63] and does not impact the catalytic properties of TEM-1[6,35]. Overall, we examined the single mutants G238S, E104K and A42G, as well as the double mutant E104K/G238S (TEM-15) and the triple mutant A42G/E104K/G238S. E104K occurs in clinical isolates (TEM-17) and is located at the 105-loop, which shares a dense interaction network with other primary substitution sites. A42G, identified in laboratory evolution experiments[29,30,36,59,64], is found at the $\alpha_1$-$\beta_1$ loop, which packs against the 270-loop. Similarly to G238S, E104K elicits large changes at the active site. These include effects on the N-terminus of helix $\alpha_2$, the SDN- and 216-loops, strands $\beta_3$ and $\beta_4$, with smaller perturbations in the Ω- and 238-loops (Supplementary Fig. 3a). However, no significant perturbations are observed for scaffold residues. In contrast, A42G produces large

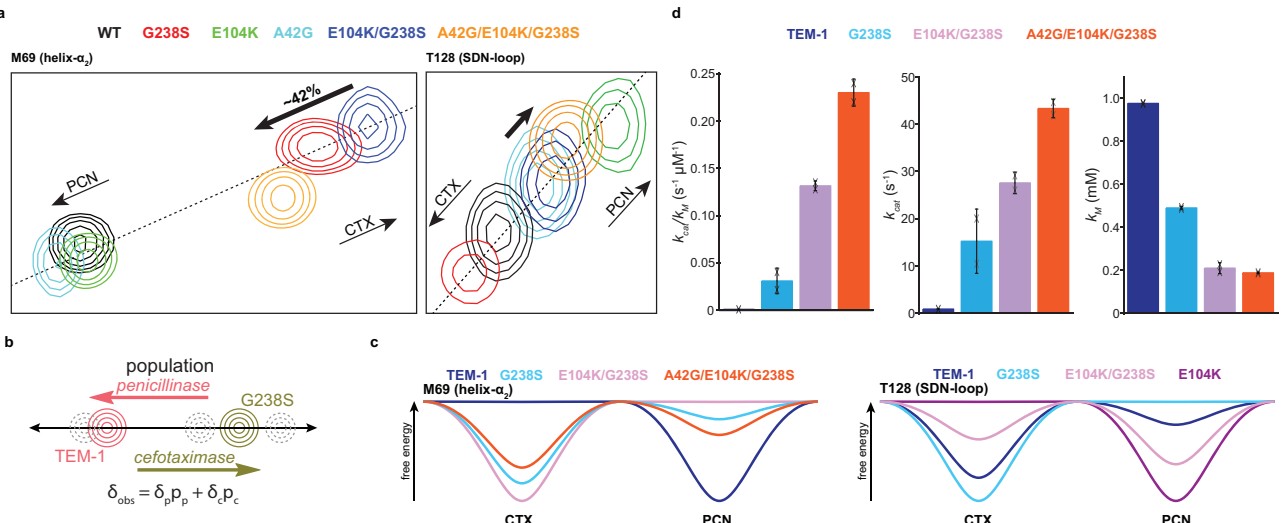

**Fig. 3 | Two-state exchange of active site walls modulates function. a** $^1$H-$^{15}$N HSQC overlays of all the evolutionary intermediates along the G238S evolutionary pathway considered in this study, showing a representative set of active site residues from helix $\alpha_2$ (M69) and SDN-loop (T128). The convention we use in the present study is that signals lying on the G238S side of the line defined by the TEM-1 and G238S signals represent a cefotaximase optimized state (CTX) and those on the TEM-1 side a penicillinase optimized state (PCN), as shown by the narrow arrows. The average population shift for the E104K/G238S → A42G/E104K/G238S step (42%) is shown by the bold arrows. The $R^2$ values for a linear fit range from 0.82 to 0.98. **b** The linear relationship observed for most structural elements indicates exchange between two states in fast exchange (microsecond to sub-microsecond). Under these conditions the observed position of a peak within the continuum (gray contours) is the population-weighted average of the chemical shift of the two states (pink and green contours), allowing for determining population shifts from changes in chemical shifts between states. **c** Representative energy landscape diagrams for two active site walls. (top) A42G causes a concerted population shift back to the penicillinase state for the E104K/G238S → A42G/E104K/G238S step, as shown here for helix $\alpha_2$. (bottom) E104K and G238S modulate the population of SDN-loop in an epistatic fashion, and in the E104K background, G238S still stabilizes the penicillinase state. **d** Kinetic parameters $k_{cat}/K_M$, $k_{cat}$ and $K_M$ for the four evolutionary intermediates along the G238S trajectory discussed in this study. Data represent the mean of two independent biological replicates with the error bar representing standard deviation.

perturbations throughout the termini of TEM-1, including the scaffold $\alpha_1/\beta_1$ and the associated loop, as well as the $\beta_5/\alpha_{12}$ region, both of which are significantly affected by the initial G238S substitution, in addition to smaller perturbations at the $\Omega$- and 238-loops (Supplementary Fig. 3b).

Examining the effect of A42G and E104K single mutants on TEM-1, together with the effects of introducing E104K in the G238S background and A42G in the E104K/G238S background to generate the second and third steps of the pathway, respectively, provides key insights on how the conformational properties of the enzyme are optimized along a model evolutionary landscape (Fig. 3a–c and Supplementary Fig. 3c). When the signal perturbations of these evolutionary intermediates are plotted together, a linear relationship is apparent (Fig. 3a, b and Supplementary Fig. 3d), irrespective of whether they belong to the active site walls or remote scaffold sites. This suggests that each of these sites exists in an equilibrium between two states that interconvert in the fast exchange regime, with the position of a signal along the line marking the population weighted average of each state (Fig. 3b). G238S, which affects all active site walls except the SDN loop, where the catalytic S130 is significantly affected (Fig. 1c and Supplementary Fig. 3d), represents the evolutionary intermediate that brings the highest increase in cefotaximase activity, with a ~100 fold higher $k_{cat}/K_M$ compared to TEM-1[6,35,44], and a significantly compromised penicillinase activity. Therefore, its relative position marks the population shift of a loop towards a cefotaximase-optimized state (Fig. 3a–c). In contrast, the relative position of TEM-1, a bona fide penicillinase, marks the population shift towards a penicillinase optimized state (Fig. 3a–c).

For E104K/G238S, the residues most affected by G238S, including the active site walls formed by the $\Omega$- and 216-loops, strand $\beta_3$ and helix $\alpha_2$, as well as scaffold sites such as the two terminal helices, $\alpha_1$ and $\alpha_{12}$, and residues of the $\alpha/\beta$ subdomain, show a chemical shift close to that of G238S in the double mutant (Fig. 3 and Supplementary Fig. 3). Thus,

even in the presence of E104K, the initial mutation utilizes a broad network of residues to modulate and control the relative population of the two states sampled by multiple TEM-1 sites (Fig. 3c). In contrast, only residues at or adjacent the SDN-loop (T128, S130 and N132) shift towards the position of E104K in the double mutant, but not other core residues like K73 at helix $\alpha_2$ (Supplementary Fig. 3d). Notably, this population shift is towards the penicillinase state (Fig. 3a and Supplementary Fig. 3d). Hence, in the G238S background, E104K utilizes a narrow residue network to control the population of only the SDN-loop conformers. Inversely, the effect of G238S on the population of the SDN loop is different in the E104K background compared to the TEM-1 background. This indicates the presence of epistatic interactions (Fig. 3c) and that the single, double and triple mutants will form a combinatorial set of cefotaximase states differing in the relative populations of active site walls and scaffold sites participating in the two-state exchange. In agreement with the critical role of the SDN-loop in substrate binding and the acylation and diacylation steps, the conformational epistasis between E104K and G238S observed for the SDN-loop correlates with a significant increase in $k_{cat}/K_M$[6,29,35,44,46] (Fig. 3d and Supplementary Table 1). When A42G is introduced in E104K/G238S, residues significantly perturbed by the single mutant, primarily in helices $\alpha_1$ and $\alpha_{12}$ and the adjacent loops, shift to the same position in the triple mutant (Supplementary Fig. 3d), indicating that A42G acts through a narrow network to exert control over the population of scaffold conformers sampled by the two terminal helices.

Of note, through a broad residue network, A42G also affects active site walls and non $\alpha_1$ and $\alpha_{12}$ scaffold residues in the vicinity of the active site, partially reversing the shift towards the penicillinase state or completely restoring the position of triple mutant signals to that of TEM-1 (Fig. 3a and Supplementary Fig. 3d). For all active site walls, including helix $\alpha_2$, strand $\beta_3$, the $\Omega$-, SDN- and 216-loops, the population shift to the penicillinase conformation is similar, and in the range of ~40-50% (Fig. 3c), suggesting that short-range cooperative

interactions within the active site lead to a concerted transition. The large subglobal population shift back to the penicillinase state caused by A42G/E104K/G238S and the local population shift of the SDN-loop caused by E104K/G238S, however, occurs despite a marked increase in catalytic efficiency for cefoxitime[35] (Fig. 3d and Supplementary Table 1). Thus, while the new function, G238S state, is essential for acquiring cefoxitimase activity, sampling a threshold level of the TEM-1 penicillinase conformation is mechanistically still required for improving $k_{cat}/K_M$. Therefore, for the G238S trajectory, secondary substitutions A42G and E104K do not modulate $k_{cat}/K_M$ by further stabilizing the G238S state, but by recovering a significant TEM-1 population, and therefore by optimizing the relative population ratio of G238S and TEM-1 states (Fig. 3c, d).

In summary, the six TEM states considered here, distinctly modulate the populations of active site walls and scaffold conformers through either broad, but subglobal, or narrow and therefore local interaction networks. This results in a combinatorial set of cefoxitimases differing in the relative population of active site wall conformers. Importantly, along the TEM-1→G238S→E104K/G238S→A42G/E104K/G238S trajectory, cefoxitimase activity is not enhanced simply by progressively increasing the population of the G238S-derived active site conformation. Instead, catalytic efficiency is optimized by fine-tuning the relative population of the original TEM-1 and the G238S active site conformations.

## Local stability changes reveal G238S-induced pleiotropic defects

While G238S shifts the population of TEM-1 active site walls to conformers that render it an efficient cefoxitimase, new-function mutations are generally destabilizing ($\Delta\Delta G > 0$), resulting in a tradeoff between activity and stability. However, given the large robustness of TEM-1 ($\Delta G_{Native}$ - $\Delta G_{Unfolded}$, $\Delta G_{N-U} > $ -7.5 kcal/mol) and the small $\Delta\Delta G$ associated with G238S ( ~ 2.0 kcal/mol)[6,35,36], a significant population of the unfolded state cannot account for the acquisition of stabilizing substitutions. Indeed, the explanatory power of $\Delta G_{n-u}$ for models that use MIC as a fitness proxy is weak, in contrast to $k_{cat}/K_M$, which drives evolution of cefoxitime resistance[35]. Moreover, G238S lowers the robustness threshold between the native state and a low-populated intermediate (2%), providing a pathway to proteolytic degradation[65].

Disturbing the system with denaturing agents or through thermal unfolding provides insight into the stability of species populated along the unfolded-native transition. However, the global nature of the derived observables does not provide information on how stability changes are distributed throughout the three-dimensional structure of a protein in its native state. To probe local stability perturbations caused by G238S at the residue level and resolve the thermodynamic contingencies imposed by an initial substitution on subsequent evolutionary steps, we employed hydrogen/deuterium-exchange coupled to mass spectrometry (HDX-MS). HDX provides a distinct advantage in that, under appropriate conditions, changes in the exchange properties between evolutionary intermediates yield direct thermodynamic information, while MS affords measurements even for amide groups that are broadened beyond detection in the NMR spectra (Fig. 1). In our analysis we measured differential deuterium uptake as a proxy of local stability changes between successive evolutionary intermediates.

For all peptides, a single isotopic envelop with continuous increase in deuterium incorporation is observed as a function of time, for both TEM-1 and G238S. This behavior is consistent with the EX2 exchange limit, where exchange at each peptide site is uncorrelated with exchange at other sites within the peptide (Supplementary Fig. 4a). Hence, differences in the deuterium uptake between evolutionary intermediates reflect variations in the thermodynamics of their corresponding local fluctuations. TEM-1 exhibits slow but high deuterium incorporation ( > 50%) after 4 h of labeling, primarily in a few exposed loops (Supplementary Fig. 4b, c), as expected for an overall stable protein. In contrast, introduction of G238S results in a

significant increase in the final level or rate of deuterium incorporation ( > 15% relative to TEM-1) across a set of peptides covering ~ 40% of the sequence (Fig. 4a and Supplementary Fig. 4b, c). Several peptides that correspond to the active site walls were identified, including 163-182 (Ω-loop), 213-218 (216-loop), 230-243 (strand β_3 and 238-loop), and 265-271 (270-loop) (Fig. 4b and Supplementary Fig. 4b–d). We note that for multiple residues in these peptides, particularly those located in loops, significant broadening is observed in the 2D $^{15}$N-HSQC of G238S (Fig. 1), indicating that acquisition of μs-ms timescale motions is accompanied by reduced local stability. Additionally, a second set of peptides, comprising long stretches of scaffold residues, displayed enhanced deuterium uptake in G238S. This includes 26-39 and 273-289, spanning the N- and C-terminal helices ($\alpha_1$ and $\alpha_{12}$), 40-46 and 244-265, covering most of the antiparallel β-sheet core (strands $\beta_1$, $\beta_4$, $\beta_5$ and $\alpha_1/\beta_1$ loop), and 220-229, following the 216-loop (helix $\alpha_{11}$ and $\alpha_{11}$-$\beta_3$ loop) (Fig. 4b and Supplementary Fig. 4b, c). The complete set of G238S peptides exhibiting reduced local stability shows spatial and sequence proximity, forming a continuous network or set of local networks that energetically couples the substitution site to remote scaffold sites (Fig. 4c).

The interface between helices $\alpha_{11}$ and $\alpha_{12}$, identified here as destabilized, along with the associated loops, forms a cryptic allosteric site, which emerges after the rotation of $\alpha_{12}$ towards $\alpha_1$, displacement of helix $\alpha_{11}$ and separation of the $\alpha_{11}/\alpha_{12}$ interface. This exposes non-polar sidechains and leaves the central β-sheet partially unprotected (Supplementary Fig. 5a)[66]. It is observed as a sparsely populated conformation in TEM-1[67–70] and is stabilized in an open state by ligands acting as allosteric inhibitors[66], which effectively disrupt the hydrophobic core of the enzyme (Fig. 5a). Two such core disrupting ligands have been identified for TEM-1[66], CBT and FTA, that each bind to two sites resulting in four binding sites (Fig. 5a): *site I*, adjacent to the orthosteric site and therefore accessible, partially disrupts the $\alpha_{11}/\alpha_{12}$ and $\beta_4/\alpha_{12}$ contacts, *sites II and III*, which are highly overlapping and require large structural rearrangements that disrupt the $\alpha_{11}/\alpha_{12}$ interface, further extending *site I* towards the core and exposing the $\beta_3$-$\beta_5$ core strands, and *site IV*, which is remote, forms along the exposed $\alpha_1/\beta_2$ interface, and is not expected to contribute in disrupting the hydrophobic core. To further explore the mechanistic consequences of G238S on the properties of TEM-1 we monitored the interaction of FTA with TEM-1 and G238S using NMR (Fig. 5b). In agreement with the HDX data, addition of FTA to TEM-1 results in only small perturbations, limited to a region that covers the orthosteric site and *site I*, indicating that the allosteric site maintains a stable hydrophobic core that is not disrupted by the ligand (Fig. 5c). Consistent with this observation, we found that TEM-1 exhibits a relatively high $K_i$ (203 ± 38 μM) for nitrocefin hydrolysis (Fig. 5d and Supplementary Fig. 5b), as previously reported[66]. For G238S on the other hand, addition of FTA causes prominent changes in chemical shifts as well as extensive line broadening to multiple signals throughout the α/β subdomain (Fig. 5b), indicating global changes, reminiscent of ligand binding at the core of the protein. Mapping this effect on the structure of TEM-1 shows that the most affected residues are found at the interface between the β-sheet ($\beta_3$-$\beta_5$) and helices $\alpha_1$, $\alpha_{11}$ and $\alpha_{12}$, suggesting that the ligand occupies *sites I* and/or *II/III*, which are only accessible when the core of the protein is partially disrupted (Fig. 5c). Therefore, in support of the HDX data, the reduced local stability observed at these structural elements for G238S allows ligand binding and partial opening of the allosteric site (Supplementary Fig. 5c, d). In agreement with this, G238S increases the potency of FTA-mediated allosteric inhibition, resulting in a 5-fold drop in $K_i$ (40.4 ± 4.0 μM) compared to TEM-1 (Fig. 5d and Supplementary Fig. 5b). Thus, while G238S optimizes the orthosteric site for efficient cefoxitime hydrolysis, it partially stabilizes the open conformation of the allosteric site that functions as an autoinhibitory element and therefore prevents the initial substitution from conferring an optimal fitness advantage.

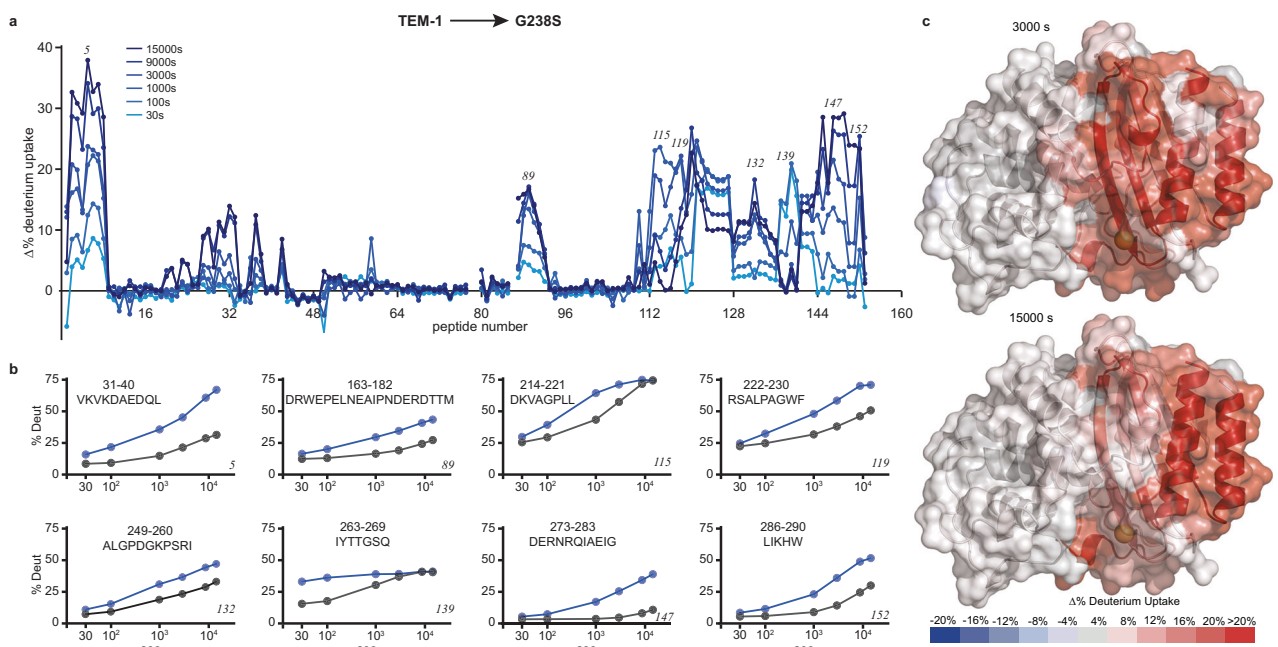

**Fig. 4 | G238S induces differential local stability defects. a** Residual plot of common peptides identified for TEM-1 and G238S expressed as the % difference of deuterium incorporation ($\Delta\%D$) between G238S and TEM-1 ($\%D_{G238S}$ - $\%D_{TEM-1}$) at different timepoints of exchange. Short exchange times are shown in light blue and long exchange times in dark blue. Numbers in italics indicate the peptide number for which uptake plots are presented in (**b**). **b** Deuterium uptake plots as a function of labeling time for a representative set of peptides showing significant difference in deuterium incorporation between G238S (blue) and TEM-1 (gray) ($\Delta\%D > 15\%$). The residue boundaries and corresponding sequence are shown on the top of each graph and the peptide number at the bottom right. **c** Differences in deuterium incorporation ($\Delta\%D$) at 3000 s (top) and 15000 s (bottom) plotted on the structure of TEM-1 highlighting how the difference in stability is disseminated at different regions of the enzyme. The position of S238 is marked with a yellow sphere.

Overall, G238S alters the equilibrium ensemble of TEM-1, enabling it to populate states with clustered distribution of local stability defects. This redistribution defines the network of energy propagation between distal sites and sets the spatial contingencies for the acquisition of secondary substitutions. Importantly, the requirement for secondary substitutions may reach beyond pleiotropic defects associated with $\Delta\Delta G$[6,35,65], to partially stabilize the network of structural elements that acquire conformations of non-native packing and have weak autoinhibitory effects[66–68] or suboptimal catalytic properties.

**Secondary substitutions restore scaffold stability while maintaining active-site dynamics**
The dynamic population shift resulting from substitutions in the G238S pathway provides insight into how these changes fine-tune the population of conformers. However, it does not reveal how the thermodynamic stability of these evolutionary intermediates is altered at the residue level throughout the pathway. To investigate this, we compared the changes in local thermodynamic stability across the evolutionary steps leading to the E104K/G238S double mutant and the A42G/E104K/G238S triple mutant. At a global level, it has been shown that E104K marginally impacts the stability of TEM-1[35]. However, when combined with G238S, E104K further decreases the $\Delta G_{N-U}$ of E104K/G238S by ~1.0 kcal/mol[6,35]. On a local level, the incorporation of E104K into the G238S background causes a significant decrease in deuterium incorporation for scaffold peptides at the termini of the lactamase fold, including the entire helix $\alpha_1$ and the C-terminal half of helix $\alpha_{12}$, as well as for peptides covering helix $\alpha_{11}$ (Supplementary Fig. 6a–c). Additionally, E104K/G238S exhibits a small increase in the deuterium uptake compared to G238S for the peptides covering the 105-loop, at the site of the first secondary substitution, and the N-terminal half of the $\Omega$-loop, including the catalytic E166 site (Supplementary Fig. 6a–c). In contrast, the exchange properties of peptides covering the SDN-loop are unaffected by the E104K substitution, suggesting that the

population shift of the SDN-loop to a cefotaximase conformation detected by NMR (Fig. 3 and Supplementary Fig. 3) is not accompanied by changes in local stability. On the other hand, A42G has been shown to have a strong impact on the stability of TEM-1 ($\Delta\Delta G_{N-U}$ ~ -3.5 kcal/mol) and restores the $\Delta G_{N-U}$ of A42G/E104K/G238S to the level of G238S[35]. Compared to E104K/G238S, the most prominent change observed upon incorporating A42G is an additional large decrease in the deuterium uptake for helix $\alpha_1$ (> 30%) (Supplementary Fig. 6d–f). This is accompanied by a further, though smaller, decrease in the uptake of the peptides covering the 216-loop and the C-terminal helix $\alpha_{12}$ (Supplementary Fig. 6d–f). Overall, the effect of the two secondary substitutions from the initial G238S step to the A42G/E104K/G238S triple mutant is additive and stabilizing (Fig. 6a, b). The scaffold regions of the TEM-1 fold that progressively show a gain in local stability form a spatially contiguous network of secondary structure elements, including helices $\alpha_1$, $\alpha_{11}$, $\alpha_{12}$, and the associated loops (Fig. 6c). The stabilizing contributions of E104K and A42G restore the stability of helix $\alpha_1$ to a level that exceeds that of TEM-1, while partially buffering the stability of helix $\alpha_{12}$ and to a lower extend that of $\alpha_{11}$ (Fig. 6b, c). Consistent with an only partial stabilizing effect, FTA can still bind to the triple mutant and inhibit its catalytic activity (Supplementary. Figure 5b). Among the peptides comprising active site elements, the $\Omega$-loop is the only region that becomes further destabilized compared to G238S and hence TEM-1. In contrast, for all other active site peptides, there is a cumulative stabilizing effect by A42G and E104K, although this does not fully compensate for the destabilizing effect of G238S. This agrees with the population shift back to the penicillinase state observed by NMR (Fig. 3), which is not complete, but occurs to the relatively more stable TEM-1 state.

In summary, the secondary substitutions E104K and A42G compensate pleiotropic scaffold stability defects caused by the initial substitutions, primarily at the terminal secondary structure elements $\alpha_1$ and $\alpha_{12}$, and helix $\alpha_{11}$. Despite this gain in scaffold local stability, the

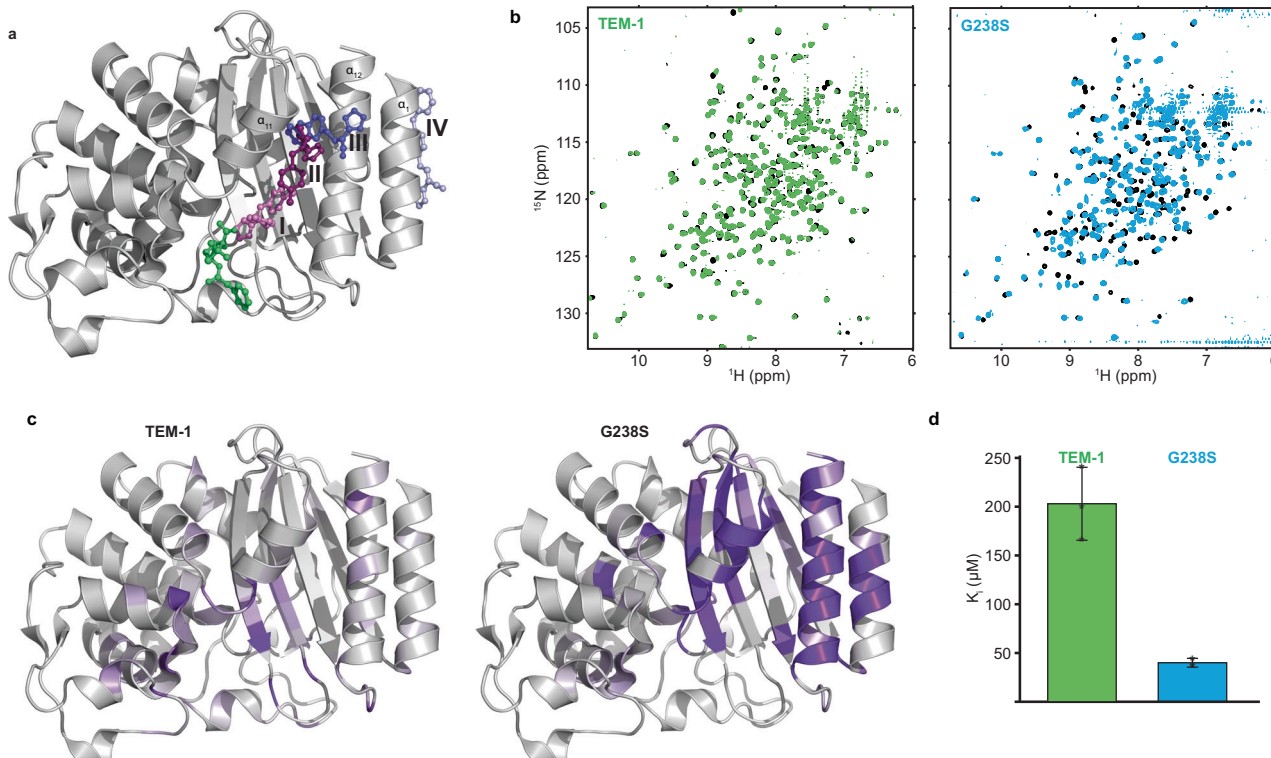

**Fig. 5 | G238S stabilizes the open allosteric site observed in the presence of weak allosteric inhibitors. a** The mode of binding for the allosteric ligands CBT (light and dark purple) and FTA (light and dark blue) illustrated on the crystal structure of TEM-1 in complex with CBT (1PZO). Marked are the four binding sites I-IV and helices α₁/α₁₁/α₁₂ that rearrange to accommodate ligand binding. Benzyl-penicillin is shown in green to highlight the position of the orthosteric site. **b** Overlay of the ¹H-¹⁵N HSQCs of TEM-1 (left) and G238S (right) in the free state (black) or in the presence of five equivalents FTA. **c** Spectral changes induced by FTA mapped on the structure of TEM-1 in complex with CBT in a light-to-dark purple gradient. For TEM-1 only small chemical shift perturbations are observed. For G238S, both chemical shift perturbations and exchange broadening are observed. Residues broadened beyond detection for G238S are colored as having a maximum effect. **d** Dose-response curves containing 10-points were measured in triplicate. Values represent the mean and error bars represent ± one standard deviation of technical replicates.

secondary substitutions either partially compensate for the stability of active site walls or even further decrease it, as is the case for Ω-loop. This suggests that the functionally relevant enhanced dynamics induced by G238S are preserved, as expected for a stabilized, but extended spectrum enzyme.

## Discussion

This iterative optimization of activity and stability is a pervasive mechanistic constraint shaping fitness landscapes, and is apparent in diverse systems, from resistance[6,71–75] to protein engineering[76–79]. While the current model captures the biophysical principles of this tradeoff, the molecular basis for most systems remains poorly understood. This limits our ability to develop predictive tools for resistance mutations, design antibiotics ahead of clinical evolution, identify alternative inhibitors to potentiate existing antibiotics, and understand how stabilizing substitutions are selected at distal sites. The search for molecular features leading to innovation often relies on static structures, disregarding the statistical ensemble nature of protein structure and therefore the impact of dynamics on protein function[80,81]. Stability defects are also described in terms of Gibbs energy of unfolding, when under native conditions this is not evenly disseminated throughout the protein structure[82–85]. Using a series of TEM-1 mutants we show that the breadth of conformations contributing to the tradeoff observed across the evolutionary landscape of β-lactamases spans an ensemble of native and non-native states that interconvert within a wide range of timescales.

Protein dynamics from picoseconds to seconds play an important role in enzyme function by controlling substrate binding, orientation,

catalytic turnover and product release[80]. Although dynamics may involve individual residues or secondary structure elements, active site loops dominate functionally significant protein motions[86]. Directed and natural enzyme evolution demonstrated that functional innovation, like the acquisition of broader specificity by TEM-1 and other β-lactamases, is accompanied by altered dynamic properties[13,54,87–92], including changes in thermodynamics and kinetics related to conformer populations and redistribution rates[80]. Extensive NMR and activity studies on TEM-1 allows dissecting the extent of dynamics required for converting it from a penicillinase to a broad-spectrum enzyme hydrolyzing both penicilins and cephalosporins (Figs. 1, 2). TEM-1 is rigid at all timescales, with few residues showing μs-ms motions[50,57,92], which prevents active site opening events that allow processing of bulky cephalosporins. cTEM-2m, an engineered TEM-1 with M68L and M69T substitutions from PSE-4, shows slow dynamics in several residues, covering a portion of the 238-loop, strand β₄ and few scaffold residues[58]. The engineered dynamics do not affect the native function, but are insufficient to expand the active site and broaden specificity, hence the enzyme remains a poor cefotaximase compared to G238S (Fig. 2). cTEM-17m, with the 150-190 segment swapped for that of PSE-4, and cTEM-19m, with both the 68-69 and 150-190 regions swapped, show enhanced dynamics at all active site walls, including the Ω-loop (Fig. 2) and the catalytic E166, which together with Lys73, activates S70 during acylation. Acylation is rate limiting for cefotaxime hydrolysis by TEM-1[93], and flexibility of E166 during this step is excessive for an efficient cefotaximase, and even when G238S is incorporated into cTEM-17m, it results in a fourfold lower $k_{cat}/K_M$ compared to G238S[94]. In contrast, the adaptive G238S in the wild-type

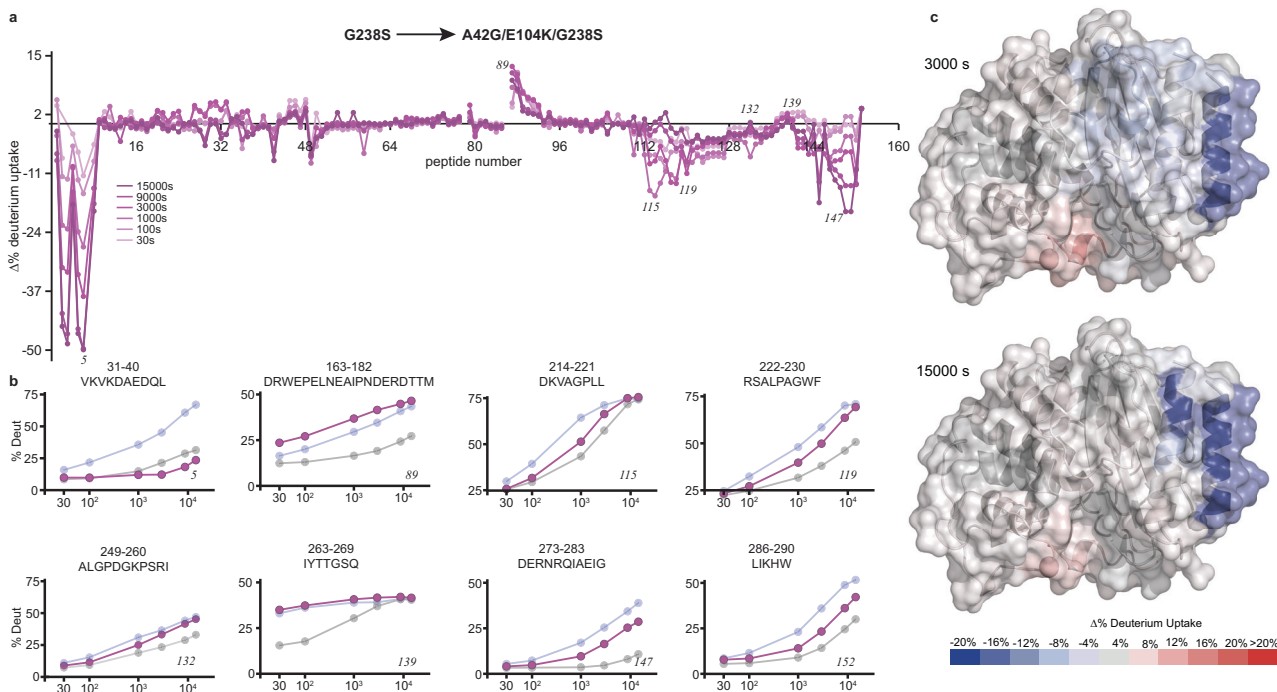

**Fig. 6 | Secondary substitutions compensate for scaffold stability defects.** **a** Residual plot expressed as the % difference of deuterium incorporation (Δ%D) between the triple mutant A42G/E104K/G238S and the initial substitution, G238S (% $D_{A42G/E104K/G238S}$ - %$D_{G238S}$) at different timepoints of exchange. Short and long exchange times are shown in light and dark pink, respectively. **b** Deuterium uptake plots as a function of labeling time for a representative set of peptides for A42G/E104K/G238S (pink), G238S (cyan) and TEM-1 (gray). The residue boundaries and corresponding sequence are shown on the top of each graph and the peptide number at the bottom right. **c** Differences in deuterium incorporation (Δ%D) at 3000 s (top) and 15000 s (bottom) plotted on the structure of TEM-1, highlighting how secondary substitutions restore stability at the scaffold elements $α_1$, $α_{11}$ and $α_{12}$. The dynamic properties of the active site are not significantly affected, except for a modest increase of the Ω-loop deuterium uptake.

background optimizes dynamics by retaining the Ω-loop rigidity and expanding the active site by enhancing flexibility in other active site walls (Figs. 1,2). This aligns with Boltzman docking studies which show a correlation between cefotaximase activity and Ω-loop rigidity[95]. Recent NMR data also demonstrate that dynamic displacement of the Ω-loop results in a suboptimal cefotaximase activity[96]. Our data therefore unravel fundamental details of functional innovation within the activity-stability tradeoff of TEM-1 evolution and highlight the delicate balance between productive and unproductive dynamics.

When examining mutants of a G238S adaptive path, it is evident that each active site wall or scaffold element undergoes a dynamic rearrangement between two states, a penicillinase and a cefotaximase optimized state (Fig. 3a, b). Along the path, the relative population change of the two states is not global but can be under the control of local networks resulting in conformational epistasis and therefore a combinatorial set of cefotaximases. G238S is at the core of a densely interconnected active site, hence it exerts the most extensive effect, perturbing the relative population of all active site walls through a broad residue network. It is a new function mutation and as such a significant population of the conformation introduced in the ensemble is preserved at all subsequent steps. However, the impact of G238S in the two-state exchange of the SDN-loop, which includes the catalytic S130 and N132, is masked in the E104K background, where through a narrow residue network, E104K determines the relative population of the exchange, shifting it towards the penicilinase state (Fig. 3a,c and Supplementary Fig. 3d). Thus, the epistatic effect between G238S and E104K accounts for the enhanced activity of the double over the single mutant. A42G on the other hand utilizes a broad but subglobal residue network to partially shift the populations of all active site wall conformers back to the TEM-1-like penicillinase state in a concerted fashion (Fig. 3a, c and Supplementary Fig. 3d). Importantly, it does so while

it still optimizes catalytic efficiency, by improving $k_{cat}$, but not $K_M$, indicating that populating the original, penicillinase conformation is mechanistically required (Fig. 3d). Therefore, evolution of cefotaximase activity by TEM-1 does not occur by progressively stabilizing the new function conformation introduced by G238S and eliminating the penicillinase state, but by optimizing the relative population of the two states. The available TEM structures provide the basis for this dynamic two-state population shift. The G238S substitution, results in a 4 Å displacement of the 238-loop and distortion of strand $β_3$ (open state), setting the original penicillinase conformation as a minor state (Fig. 3). The new function, open $β_3$/238-loop conformation of G238S and E104K/G238S (Fig. 1e) enlarges the active-site pocket, reducing the steric clash with the oxyimino substituent of cefotaxime. This results to a significantly improved $K_M$ (Fig. 3d and Supplementary Table 1), while it apparently positions the β-lactam ring more favorably to Ser70 to increase the acylation rate[93] and thus improve $k_{cat}$ (Fig. 3 and Supplementary Table 1). Since acylation remains rate limiting, the population shift to the original state and the associated enhanced $k_{cat}$ observed for the E104K/G238S → A42G/E104K/G238S step[35] (Fig. 3 and Supplementary Table 1) is a result of optimizing the deacylation rate, $k_3$. Therefore, there is a trade-off between substrate positioning for binding and acylation, and catalytic pre-organization required for deacylation. Indeed, N170, coordinating the deacylating water, samples a catalytically incompetent state in the presence of G238S[46], which agrees with the exchange broadening observed for the C-terminus of the Ω-loop (Fig. 1c). Our results suggest that the population distribution of the conformers is critical. The ensemble must include a threshold fraction of the binding and acylation competent open conformation to confer a new function, yet a sufficient fraction of the deacylation competent, original conformation is still required. Then, it is the dynamic population shift that allows high turnover of cefotaxime

of the final evolved state. Hence, selection favors mutations that cause population shifts rather than exclusively stabilizing one conformation.

The iterative two-state population shift of active-site walls provides insights into the activity arm of the tradeoff, but, evolutionary intermediates introduce pleiotropic effects that influence fitness through changes in protein stability. Indeed, G238S initiates cefotaxime resistance but reshapes the folding energy landscape by destabilizing the native state[6,35,65]. To date, the activity stability tradeoff is accounted for by changes in global thermodynamic stability, expressed as the free energy difference between the fully unfolded (U) and native (N) states of a protein ($\Delta G_{N-U}$). However, due to the lack of global cooperativity under native conditions, not all residues are affected by mutations and only a subset participates in signal propagation[83,97,98]. This network of residues is dense and involves multiple remote scaffold sites when active site residues are mutated[99], is not intrinsically bidirectional[83,100], is rich in low stability residues[83,101], it can be linked through dynamic or structural changes[102], and can be contiguous or disperse[100]. Identifying residues comprising locally unstable regions is critical for understanding how secondary mutations are fixed in response to initial active site mutations and therefore for explaining how remote sites are selected.

Local stability refers to the dynamic, site-specific conformational fluctuations that occur within the native state ensemble and can be very accurately measured using HDX or chemical footprinting. These are not large-scale unfolding events, but rather, localized dynamics that do not cause loss of the overall fold or catalytic activity. Often result in response to selection pressure and optimize enzyme function, such as enhanced local dynamics that improve $k_{cat}/K_M$ or $K_d$, providing a significant fitness advantage. For example, it was found that four evolutionary related CTX-M β-lactamases, show no difference in their structure[103], yet their distinct local stability profiles correlate with improved catalytic efficiency[54,103]. In addition, local stability can be of pleiotropic origin and result in folding defects by disturbing the equilibrium between locally protected (closed) and exposed (open) states. These subtle deviations can open a pathway toward misfolding, aggregation and degradation as it has been previously shown for TEM-1[65]. Other pleiotropic folding defects include changes in kinetic stability[38], as well as altered in vivo properties related to crowding and interactions with other cellular partners. Similarly to global stability, local defects may be corrected in subsequent evolutionary steps through broad or narrow networks.

To map the residue networks involved in the energetic coupling between initial active site mutations and secondary mutations along the cefotaximase evolution of TEM-1, we used HDX. Measuring differential deuterium uptake as a proxy of local thermodynamic stability allowed us to identify the spatial contingencies in energy propagation imposed by G238S and probe changes in local stability between other evolutionary intermediates. G238S does not affect only active site residues, but increases the level or rate of deuterium incorporation in remote scaffold peptides spanning helices $\alpha_1$, $\alpha_{11}$ and $\alpha_{12}$, and the core β-sheet. These peptides share spatial and sequence proximity, forming the broad network or a set of local networks that energetically couples G238S to remote sites (Fig. 4c). In addition, the $\alpha_{11}$-$\alpha_{12}$ interface of TEM-1 forms a transient allosteric pocket of non-native packing, where the hydrophobic core becomes partially exposed (Supplementary Fig. 5). A42G, which is part of the $\alpha_1$/$\alpha_{11}$/$\alpha_{12}$ network, and to a lower extent E104K, compensates for these local scaffold defects without compromising the beneficial effects of G238S[83,100], by restoring the stability of helices $\alpha_1$ and $\alpha_{12}$, and less significantly that of helix $\alpha_{11}$ (Fig. 6 and Supplementary Fig. 6). Hence, this explains why a A42G is fixed at this remote scaffold site in the G238S background and highlights how coupled dynamics propagate long-range coupling and stability defects[39–41].

Our results reinforce the critical role of $\alpha_1$/$\alpha_{11}$/$\alpha_{12}$ network stability on the folding properties of TEM-1. N-terminal deletion mutants exhibit significantly lower stability and impaired in vivo activity at 37 °C[104], consistent with native states stabilized by N-/C-terminus interactions[104]. TEM-1 folds through a stable thermodynamic intermediate[105] (*I*), and stability loss in these deletion mutants is attributed to a lower $\Delta G_{N-I}$, rather than $\Delta G_{I-U}$, suggesting that disrupting the $\alpha_1$/$\alpha_{11}$/$\alpha_{12}$ network increases the population of *I*. When these constructs are subjected to directed evolution, A42G is selected as a stabilizing substitution[104], supporting our findings on the role of helix $\alpha_1$ in the stability of the $\alpha_1$/$\alpha_{11}$/$\alpha_{12}$ network and in maintaining a native state. Similarly, W290 mutations, at the C-terminus of helix $\alpha_{12}$, cause deleterious folding defects, and even the conservative W290F substitution results in a prominent decrease in $\Delta G_{N-I}$ and a tenfold increase in the unfolding rates[106]. In agreement with these data, the network can also be stabilized by engineering a cyclic TEM-1[107,108] or by A224V, a mutation on $\alpha_{11}$, at the interface with $\alpha_{12}$, that has been identified multiple times in laboratory evolution experiments and clinical isolates as a secondary substitution. Finaly, it was recently shown using HDX, that local stability defects modulate the in-cell kinetic stability profiles of NDM metallo-β-lactamases. $Zn^{2+}$ binding or naturally occurring variants alter the dynamics of a long C-terminal segment resulting in differential access to proteolytic degradation[109].

Despite the distinct dynamic and local stability demands of unrelated proteins, the mechanistic insights into the activity-stability tradeoff of TEM-1 evolution provide a valuable framework for understanding the tradeoff in other systems. Our results show that adaptive trajectories exploit conformational heterogeneity to achieve innovation without sacrificing robustness. Mutations do not act as binary switches between structures but as rheostats that reconfigure the ensemble to optimize different mechanistic steps. Similalry, crystal structures in the presence of engineered or acquired secondary stabilizing substitutions may accurately recapitulate active site geometry but may also mask subtle scaffold defects that affect both stability and activity[46]. Mapping the network of energetic coupling to initial mutations using HDX is expected to aid in improving industrial enzyme performance and developing effective therapeutic proteins. These networks indicate regions where compensatory mutations can be introduced, narrowing the sequence space and increasing the efficiency and success rate of protein engineering efforts[110].

## Methods

### Sample preparations

TEM-1 was expressed from a modified pET-GST vector using its signal sequence for secretion to the periplasmic place, yielding a sample with a native N-terminal sequence. The same TEM-1 construct was used as a template for mutagenesis using the QuikChange Lightning Site Directed Mutagenesis Kit to prepare G238S, R164K, E104K, A42G, E104K/G238S and A42G/E104K/G238S. Mutations were confirmed by sequencing. Protein expression was performed in E. Coli BL21(DE3) grown in LB media for preparing unlabeled samples for HDX and kinetic studies. For the preparation of isotopically labeled samples used for NMR analysis TEM-1 proteins were expressed in M9 minimal media supplemented with either $^{15}NH_4Cl$ or $^{15}NH_4Cl$/$^{13}C$-glucose for preparing $^{15}N$- and $^{13}C$/$^{15}N$-labeled samples, respectively. Cells were grown in the presence of 50 µg/mL kanamycin and incubated at 37 °C to an $OD_{600}$ ~ 0.6. Expression was induced with 0.5 mM IPTG at 22 °C for 18 h. Cell cultures were centrifuged at 4 °C at 5000× *g* for 30 minutes and pellets were resuspended in 25 mM Tris-HCl pH 8.4. Resuspended cells were lysed by sonication with 1 mM PMSF and 0.1 µM of each of Leupeptin, ABSF, Bestatin, and Pepstatin protease inhibitors. Lysed cells were centrifuged for 1 h at 35,000× *g*. The supernatant was diluted and loaded on an anion exchange column (Q Sepharose) equilibrated with 25 mM Tris-HCl pH 8.4 and eluted using an isocratic gradient of 0-

100 mM NaCl, followed by a wash with 100 mM NaCl and a second isocratic gradient of 100 - 500 mM NaCl. TEM-1 containing fractions were pooled, exchanged to 25 mM Tris-HCl pH 8.4, and further purified through a second anion exchange column (HiTrap Q HP 5 mL) run with the same elution scheme as the Q Sepharose. TEM-1 fractions were concentrated to 5 mL using a stirred cell (10 kDa cutoff) and further purified via a size exclusion column (HiLoad 26/600 SuperDex 200 prep grade) in 25 mM Tris-HCl pH 8.4, 150 mM NaCl. Purity of samples was confirmed by SDS-PAGE. Samples were aliquoted, snap-frozen in liquid nitrogen and stored in -80 °C.

## NMR spectroscopy

All NMR experiments were performed in 25 mM PIPES pH 6.4 with 7.5% D2O, at 30 °C and at a concentration ranging from 0.4 to 0.8 mM. The available TEM-1 assignment[50] was used to assign single mutants of the G238S pathway based on signal proximity and was further aided by a 3D HNCA/HN(CO)CA pair of triple resonance spectra acquired on a Varian direct drive 800-MHz instrument equipped with a cryoprobe. The double and triple mutants were assigned by transferring the assignment from the single mutants and aided by an HNCA/HN(CO)CA pair. For G238S only, a $^{15}$N-edited 3D NOESY was also acquired. Backbone $^{15}$N $R_1$ and $R_2$ relaxation rates, $^{15}$N-($^1$H) heteronuclear NOEs and $^{15}$N CPMG relaxation dispersion experiments were acquired on Bruker Avance Neo 600 and Avance III 850 MHz at 30 °C with $^{15}$N-HSQC-based interleaved pseudo-3Ds. $^{15}$N $R_1$ rates were measured using the standard Bruker pulse sequence and relaxation delays of 0.025, 0.05, 0.10, 0.20 (x2), 0.30, 0.40, 0.50, 0.60, 0.75, 1.00 (x2), 1.25, 1.50 and 1.75 s. $^{15}$N $R_2$ rates were measured using a CPMG length of 16.5 ms and relaxation delays of 0.017, 0.034, 0.051, 0.068, 0.102 (x2), 0.136, 0.170, 0.204, 0.237 and 0.305 s. $^{15}$N-hNOEs were obtained from signal intensities of spectra acquired in the presence and absence of a 4-s proton saturation pulse, and errors were calculated from the S/N of the spectra. The exchange contribution to the transverse relaxation rate ($R_{2,eff}$) was determined using a modified CPMG sequence that reduces off-resonance effects[111], with a total CPMG time of 40 ms and pulse frequencies of 0, 50, 100 (x2), 150, 200, 250, 300, 350, 400, 450, 500 (x2), 600, 700, 800, 900, 1000 (x2), 1200, 1600 and 2000 s$^{-1}$. All spectra were processed using NMRpipe[112] and analyzed using sparky (University of California, San Francisco).

Backbone $^{15}$N $R_1$, $R_2$ and hNOEs were interpreted with the model-free formalism[113] using scripts integrated in relax[114,115]. The D'Auvergne protocol was used[114], which optimizes the model-free parameters without global parameters and then refines the diffusion tensor models, mitigating issues related to the initial tensor estimate. Briefly, for tensor determination spins are excluded based on three criteria: (1) lack of clear secondary structure, as determined by the DSSP algorithm, (2) spectral overlap, and (3) NOE values lower than 0.8. The optimal tensor is selected using the Akaike Information Criterion (AIC)[116,117]. Once the diffusion model is established, multiple model-free models (m1 to m5) are fitted and optimized for each spin, except those exhibiting overlap. The best model was selected based on AIC. CPMG data were analyzed using relax. When relaxation dispersion is identified ($\Delta R_{2,eff} > 4$ s$^{-1}$) curves were fitted to the Luz–Meiboom[118] or Carver–Richards[119] models for extracting $k_{ex}$ or $k_{ex}$, $\Delta\delta$ and populations. AIC was used for model selection. Errors in both model-free and relaxation dispersion fitting were determined via 500 Monte Carlo simulations in relax. Chemical shift perturbations were calculated as $^1$H-$^{15}$N combined chemical shift, $\Delta\delta = \sqrt{\Delta\delta_H^2 + \left(\frac{\Delta\delta_N}{5}\right)^2}$.

The TEM-1 and G238S sample concentration used to monitor FTA binding was 0.2 mM, while the FTA concentration was 1.2 mM. Both reference and FTA-bound spectra were acquired in the presence of 4% DMSO to keep the ligand soluble.

## Hydrogen/deuterium exchange

HDX experiments were initiated by diluting TEM-1 and its mutants from stock solutions prepared in 25 mM MES, pH = 6.4 and H2O (60-100 μM), 11-fold into buffer prepared in D2O (pD = 7.1) at 15 °C. At a designated exchange time (30, 100, 1000, 3000, 9000, and 15000 sec), the exchange was quenched by diluting the samples with an equal volume solution of cold quenching solution (0.8% formic acid, 3.0% acetonitrile in H2O). The mixture was injected by the LEAP HDX liquid handling robotics system to Enzymate BEH Pepsin column (Waters) for digestion and further analysis. The digestion was performed at a flow rate of 0.15 ml/min using 0.15% formic acid/3% acetonitrile as the mobile phase. The resulting peptides were collected and desalted with an inline 4-μl C18- Opti-lynx II trap cartridge (Optimize Technologies) and then eluted through a C-18 column (Thermo Fisher Scientific, 50 × 1 mm Hypersil Gold C-18) using a rapid gradient from 2 to 90% acetonitrile containing 0.15% formic acid and a flow rate of 0.04 ml/min, leading directly into a maXis-II ETD ESI-QqTOF mass spectrometer. The total time for the digest and desalting was 3 min, and all peptides had eluted from the C-18 column by 15 min. The peptide fragments were identified using Bruker Compass and Biotools software packages. The level of deuterium incorporation was assessed using the commercial software HDExaminer-3 (Trajan Scientific). The standard deviation for deuterium incorporation from triplicate technical measurements at the representative time points of 30 and 300 seconds did not exceed 0.15 Dalton. Experiments were performed in the absence of any reducing agent to maintain the native disulfide bond between C77 and C123. Representative peptides that included the two cysteines were identified during the analysis and confirmed that no exchange occurred in this region of the proteins.

Constructs used in this study produce periplasmic TEM proteins with native N-terminus starting at position H26. Residue numbering is according to the Ambler scheme for class A β-lactamases, which places the active site serine at number 70. For TEM-1, this numbering includes gaps between residues G238-E240 and P252-D254, therefore positions 239 and 253 are not included in the sequence.

The HDX summary table is included in the supplementary information (Supplementary Table 3).

## Enzymatic assays

The catalytic activity of TEM-1 and mutants was measured using cefotaxime and nitrocefin as substrates. Cefotaxime hydrolysis was monitored at 260 nm using a Biotek Cytation multimode plate reader at 25 °C in 50 mM sodium phosphate buffer (pH 7.0). FTA inhibition was assessed using serial dilutions of inhibitor (starting at 1.2 mM, 800 μM, or 500 μM), with cefotaxime concentration ranging from 500 to 100 μM, and enzyme concentrations between 1 μM and 2 nM, depending on catalytic activity. Nitrocefin hydrolysis was measured at 486 nm at 37 °C in 50 mM Tris buffer (pH 7.0), using inhibitor dilutions from 800 μM, with 100 μM to 10 μM nitrocefin and 0.5 nM enzyme.

Kinetic parameters ($V_{max}$ and $K_M$) for cefotaxime and nitrocefin were determined by nonlinear regression fitting to the Michaelis-Menten equation using SigmaPlot 12.5. IC$_{50}$ values were obtained from sigmoidal concentration response curves, and $K_i$ values were calculated using the Cheng-Prusoff equation: $K_i = IC_{50}/(1 + [S]/K_M)$, where [S] is the substrate concentration (cefotaxime or nitrocefin), and $K_M$ is the Michaelis constant of each enzyme.

## Reporting summary

Further information on research design is available in the Nature Portfolio Reporting Summary linked to this article.

## Data availability

The HDX-MS data generated in this study have been deposited in the ProteomeXchange Consortium via the PRIDE partner repository under

accession code PXD069558. Previously published BMRB codes utilized: BMRB 7236. Source data are provided in the source data file. Materials used are available from the corresponding author.

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

## Acknowledgements

We are grateful to Dr. Youlin Xia (St. Jude Children's Reseach Hospital) for providing the CPMG pulse sequences, Dr. Todd Rappe (Minnesota NMR Center) for assistance with the relaxation experiments and Sofia Gonzalez for assistance with sample preparations. NMR experiments were carried out at USF's Florida Center of Excellence for Drug Discovery and Innovation and the Minnesota NMR Center. This work was supported by the US National Institutes of Health (GM115854 to I.G. and AI161762 to Y.C.).

## Author contributions

E.A., D.K., Y.C. and I.G. designed research; E.A., D.K. and M.P. prepared samples; E.A., D.K., V.K. and J.M.D. acquired and analyzed NMR data; E.A., R.A. and I.G. analyzed HDX-MS data; L.J. performed enzymatic assays; S.M., Y.C. and I.G. provided resources; D.K. and I.G. drafted the manuscript; E.A., D.K., M.P., J.M.D., Y.C. and I.G. reviewed and edited the manuscript.

## Competing interests

The authors declare no competing interests.
