## [Transparent Peer Review file · Nature Communications]

Dynamic signature of activity-stability tradeoff in lactamase evolution

Corresponding Author: Dr Ioannis Gelis

Version 0:

Reviewer comments:

Reviewer #1

(Remarks to the Author)

The manuscript by Arcia and coworkers provides a compelling and detailed NMR-based study of the dynamic features in a key evolutionary pathway of the β -lactamase TEM-1 towards cefotaxime resistance. This enzyme has been thoroughly studied as a model for protein evolution. In this regard, the authors rely on a large amount of clinical evidence, biochemical and structural studies over three decades. The initial substitution, G238S, induces chemical shift perturbations all over the structure. CPMG relaxation dispersion experiments on this variant can be described by a two-state model, that the authors interpret as the two conformations adopted by the 238-loop upon this mutation (open and closed) in fast exchange in the μ -ms timescale. Instead, R164S elicits an exchange phenomenon slow in the NMR timescale, as witnessed by the duplication of several signals. Other mutants in the evolutionary pathway TEM-1 \rightarrow G238S \rightarrow E104K \rightarrow A42G \rightarrow M182T are analyzed in terms of chemical shift perturbations. The authors conclude that a linear relationship of the CSPs with these perturbations reveal that these variants are sampling two extreme conformations, one favoring hydrolysis of penicillin, and the evolved one, favoring cefotaxime hydrolysis. Then, they explore stability perturbations by HDX-MS, disclosing the pleiotropic effects of mutation G238S. Finally, they assess the compensatory role of mutations E104K and A42G.

Overall, this manuscript assesses directly the role of protein dynamics in the evolution of a well-studied system, mostly by NMR spectroscopy. One main challenge of this manuscript is to convince the readership that the authors are not “reinventing the wheel”. In this regard, I am convinced that the information is novel, but there are some sections in the manuscript that heavily rely in already published information. The authors should better work some of these sections.

There are some specific aspects that the authors could address to further improve the quality of their contribution:

1. The authors only report CPMG experiments for the G238S variants, while the rest of the interpretation mostly relies in CSP analysis. Given that the main conclusions of the paper are regarding protein dynamics, some results should be supported by additional relaxation measurements, or the impact of temperature in the proposed equilibria.
2. Several papers have shown that the evolutionary pathway here studied is not only dominated by an activity-stability tradeoff, but by strong epistatic interactions (see papers from Weinreich, Palzkill and Shoichet cited). There are no comments on this regard. The authors have studied different combinations of mutations that can reflect the impact of epistatic effects (for example, mutation G238S in the wt background, versus the E104K background).
3. The authors conclude that “In summary, mutations that define the evolution of TEM-1 to TEM-1* do not cause a global population shift between two unique conformations, a penicillinase and a cefotaximase. Instead, mutations independently optimize the population of different regions, including active site walls and scaffold sites, resulting in a combinatorial set of TEM-1 conformers from which a potent cefotaximase conformation is optimized along the pathway” Could this be interpreted as reflecting epistatic interactions?
4. The manuscript clearly describes the dynamic effects of individual mutations, and relies in a large amount of previously published activity and MIC data. A recompilation of this information from the literature in a Supplementary table to illustrate quantitatively the impact of each mutation in activity and fitness would help the reader who is not acquainted with the details of evolution of TEM lactamases. Additionally, if feasible, including a column summarizing the overall dynamic impact of each mutation (e.g., $\Delta\Delta G$). The authors choose MS instead of NMR for studying the impact of HXD based on the fact that MS

affords measurements even for amide groups broadened due to exchange phenomena. NMR was used to interrogate the dynamics of lactamases from the CTX-M family. Did they give it a try?

Reviewer #2

(Remarks to the Author)

Arcia et al. present a comprehensive study on the role of protein dynamics in the activity-stability tradeoff of TEM-1 β -lactamase and its evolutionary adaptations toward cefotaxime resistance. Employing a combination of NMR, HDX, the authors systematically dissect the dynamic behavior of several TEM-1 variants, focusing on key mutations (G238S, E104K, A42G) and the combined effects of these mutations along the evolutionary trajectory. This approach allows for a detailed assessment of how primary and secondary mutations differentially modulate dynamics and local stability.

The study emphasizes the significance of local and global dynamics in shaping the functional landscape of TEM-1 and identifies the mutations propagate to modulate many sites around active site wall as well as remote scaffold sites. While the study provides a methodologically robust analysis of TEM-1 dynamics and the evolutionary tradeoff between activity and local stability, emphasizing the conceptual novelty and organizing key findings more clearly would significantly enhance its impact on the fields of protein engineering and antibiotic resistance research.

It would be beneficial for the authors to address the following concerns and provide clarifications prior to publication:

1. The authors extensively analyze the dynamics of single and combined mutations along the evolutionary trajectory. The Weinreich group (Ref. 35) characterized 16 variants comprising all combinations of the four mutations and discussed pleiotropic effects and epistasis, measuring catalytic efficiency and thermostability (ΔG of folding). They concluded that cefotaxime resistance in TEM-1 β -lactamase is primarily driven by enhancements in catalytic activity rather than changes in thermostability. Furthermore, they noted that subsequent mutations primarily reinforce local stability or kinetic stability rather than global stability. The manuscript uses the terms local stability and global thermodynamic stability interchangeably, which could be misleading. The authors should clearly define these terms and distinguish their roles in pleiotropy and evolutionary compensation. Additionally, clarifying how acquired mutations specifically compensate for deleterious effects would align the current findings with prior work and highlight the novel contributions of the study.
2. Studies by Bowman et al. using thiol labeling and NMR addressed the dynamic interplay between closed and open conformations of the Ω -loop in β -lactamases. Beta-lactamases with a higher population of the open Ω -loop conformation exhibit increased catalytic efficiency against benzylpenicillin, a smaller β -lactam substrate, whereas the closed conformation is more favorable for the hydrolysis of larger substrates like cefotaxime. The findings in Bowman et al. appear to align with the current work, yet the manuscript does not explicitly connect these studies. Incorporating a discussion of how the observed dynamic shifts in the Ω -loop correlate with substrate specificity in the current work would strengthen the contextual framework and provide a more integrated perspective.
3. The linear chemical shift analysis presented in Figure 4, illustrating the introduction of E104K in the G238S background and A42G in the E104K/G238S background, is compelling. However, the text is somewhat ambiguous. While the authors state that mutations independently optimize distinct regions without inducing a global population shift between cefotaximase- and penicillinase-optimized conformations, the analysis simultaneously implies that such states exist. This dual framing is conceptually inconsistent. It would be beneficial to clearly define what constitutes a cefotaximase-optimized conformation versus a penicillinase-optimized conformation. Additionally, providing quantitative data, such as interatomic distances, dihedral angles, or root-mean-square fluctuations, would concretely delineate how these states differ. Further, a clarification is needed on whether the observed shifts represent distinct, discrete states or a continuous ensemble of conformations. Statistical validation of the linearity of the chemical shift data should also be provided to ensure that the observed trend is not an artifact of selective data reporting or overfitting.
4. The authors effectively utilize HDX data to assess how G238S induces local destabilization and how E104K and A42G subsequently mitigate these effects. However, the manuscript predominantly discusses changes in folding stability ($\Delta\Delta G$ values), which do not directly correlate with HDX data reflecting localized stability perturbations. The manuscript would be strengthened by integrating additional analyses that specifically quantify changes in local stability, potentially including entropy-enthalpy compensation to capture how different regions are variably affected by each mutation. Additionally, molecular dynamics simulations, as reported in Yu et al. (Proc. Natl. Acad. Sci. USA 2018) and Ose et al. (Biophys. J. 2023), provide supporting evidence for the findings in Figures 5 and 6. Citing these studies and discussing how they align with the current work would provide a more comprehensive contextual framework.
5. The identification of a cryptic allosteric site formed by the $\alpha 11$ - $\alpha 12$ interface is briefly mentioned but not sufficiently analyzed mechanistically. The cited works also do not provide a clear mechanistic explanation. How this site influences catalytic efficiency and inhibition remains unclear. Is the formation of this cryptic site a compensatory mechanism or an unintended structural consequence of the mutations? More direct evidence is needed to substantiate the proposed role of this site, potentially including data on how ligand binding, mutational analysis, or dynamic perturbations influence its functional role.
6. The authors assert that their findings offer actionable strategies for enhancing enzyme function while mitigating stability tradeoffs. However, the broader applicability of this framework beyond TEM-1 is not clearly articulated. Explicitly discussing how the dynamic tradeoff framework developed here could be applied to other enzymes, particularly those with known allosteric networks or cryptic binding sites, would broaden the impact of the study and reinforce its relevance to protein engineering and evolutionary biology.

Reviewer #3

(Remarks to the Author)

Review for Arcia et al.,
“Dynamic signature of activity-stability tradeoff in lactamase evolution”

Arcia et al. describe how mutations in the early evolution of antibiotic resistance in TEM-1 beta lactamase influence the protein's substrate specificity and overall conformational ensemble properties. They use a combination of NMR spectroscopy and HDX/MS to achieve high spatiotemporal resolution in their study of the allosteric effects in the protein ensemble.

It's a very interesting paper, but I focus here only on the HDX/MS data analysis. There are some errors that affect the interpretation of the data. Some of them might come from errors in offsetting the residue indexing in the analysis. One would assume that the authors meant to maintain indexing that places G238S at position 238, but the indexing is inconsistent no matter what the reference is. Apart from the ambiguity in the annotation, in several cases the authors actually have the wrong number of residues in a peptide, which means that their reported %D is very likely incorrect, if the data themselves are correct. Additionally, this means that some of the peptides are not possible to correctly identify, because if the residue numbering is correct, then the data would correspond to the reported peptide, but if the residue numbering is off by -3 to +3 as in some of the other peptides, then the reported data correspond to a different peptide altogether.

In Fig. 5b:

- Peptide 32-40 should be 31-40: VKVKDAEDQL
- Peptide 163-182: DRWEPELNEAIPNDERDTTM is correct
- Peptide 211-221 should be 214-221: DKVAGPLL
- Peptide 222-230: RSALPAGW is correct
- Peptide 230-248 is ambiguous: it is possible that it should be 230-247: FIADKSGAGERGSRGIIA given the indexing problem, or alternatively, it might be correctly identified as 230-248: FIADKSGASERGSRGIIAA (it's not clear which peptide these data correspond to)
- Peptide 263-269 should be 261-267: IYTTGSQ
- Peptide 273-283 should be 271-281: DERNRQIAEIG
- Peptide 286-290 should be 284-288: LIKHW

Also in Fig. 5c, the authors might want to add the yellow sphere mentioned in the caption to the figure.

There are also mistakes in Fig. 6b:

- Peptide 32-40 should be 31-40
- Peptide 163-182 is correct
- Peptide 211-221 should be 214-221
- Peptide 222-230 is correct
- Peptide 230-248 is ambiguous: if the peptide sequence is identified correctly, then it should be 230-247, but if the indexing is correct, it should likely correspond to peptide 230-248: FIADKSGAGERGSRGIIAA (also, it is not proper to compare the deuterium uptake for two peptides where there is a difference in the sequence on the same graph, because the k_{int} for the mutated residue will be different and we cannot account for the magnitude of this difference in the %D – the authors should really find another way to show the data for these peptides)
- Peptide 263-269 should be 261-267
- Peptide 273-283 should be 271-281
- Peptide 286-290 should be 284-288.

Did not look at the supplement but the authors should double-check all their peptides.

Version 1:

Reviewer comments:

Reviewer #1

(Remarks to the Author)

The authors have adequately addressed all my concerns, providing new experiments, and appropriately explaining the experimental limitations to perform some assays.

Overall, this resulted in a great manuscript that will be impactful in the field.

Reviewer #2

(Remarks to the Author)

The authors addressed all the questions I raised.

Reviewer #3

(Remarks to the Author)

Thank you to the authors for correcting the mistakes in their uptake plot annotations. But the authors still did not correct the uptake plots showing peptides of different sequences together on the same axes. This is incredibly improper, and there are a

lot of problems with their handwaving explanation about why this is OK - it is not OK, and it sets a bad precedent in the literature for our field. The biggest problems with their explanation: 1) mutations affect both N- and C-terminal residues' exchange rates (see recent work from Hamuro showing this in a tabular format, JASMS); 2) even single mutations often change the local/global structure of the protein, which commonly manifest as changes in forward AND back exchange. The authors do not take back-exchange into account in this work. So it would be great if the authors could please treat the data properly and put the uptake for peptides of different sequences in different plots. Thanks.

Point-to-point response to reviewers' comments.

Reviewer #1: *The manuscript by Arcia and coworkers provides a compelling and detailed NMR-based study of the dynamic features in a key evolutionary pathway of the β -lactamase TEM-1 towards cefotaxime resistance. This enzyme has been thoroughly studied as a model for protein evolution. In this regard, the authors rely on a large amount of clinical evidence, biochemical and structural studies over three decades. The initial substitution, G238S, induces chemical shift perturbations all over the structure. CPMG relaxation dispersion experiments on ~~this variant can~~ be described by a two-state model, that the authors interpret as the two conformations adopted by the 238-loop upon this mutation (open and closed) in fast exchange in the μ s-ms timescale. Instead, R164S elicits an exchange phenomenon slow in the NMR timescale, as witnessed by the duplication of several signals. Other mutants in the evolutionary pathway TEM-1 \rightarrow G238S \rightarrow E104K \rightarrow A42G \rightarrow M182T are analyzed in terms of chemical shift perturbations. The authors conclude that a linear relationship of the CSPs with these perturbations reveal that these variants are sampling two extreme conformations, one favoring hydrolysis of penicillin, and the evolved one, favoring cefotaxime hydrolysis. Then, they explore stability perturbations by HDX-MS, disclosing the pleiotropic effects of mutation G238S. Finally, they assess the compensatory role of mutations E104K and A42G.*

Overall, this manuscript assesses directly the role of protein dynamics in the evolution of a well-studied system, mostly by NMR spectroscopy. One main challenge of this manuscript is to convince the readership that the authors are not "reinventing the wheel". In this regard, I am convinced that the information is novel, but there are some sections in the manuscript that heavily rely in already published information. The authors should better work some of these sections.

There are some specific aspects that the authors could address to further improve the quality of their contribution:

1. The authors only report CPMG experiments for the G238S variants, while the rest of the interpretation mostly relies in CSP analysis. Given that the main conclusions of the paper are regarding protein dynamics, some results should be supported by additional relaxation measurements, or the impact of temperature in the proposed equilibria.

Response: We fully agree that this approach would provide robust evidence, but we would like to clarify the technical challenges we encountered. The primary challenge in collecting a full set of CPMG data for all mutants was the profound line broadening for the G238S-containing mutants. Many resonances are broadened beyond detection, while others are broadened to the extent that reliable fitting and extraction of relaxation parameters is not possible. As detailed in Supplementary Table 1, even residues that did show a dispersion curve -such as

S130, V216, K234, and D254- yield poor fits, making the data unreliable for quantitative analysis (e.g., see the dispersion curve for residue L220 in Fig. 1, below).

Given these limitations, we relied on the analysis of chemical shifts, which are very sensitive to conformational exchange. Our observation of a strong linear correlation in the signal position across the series of TEM mutants provides compelling evidence for a two-state exchange process. The agreement between CPMG and CSP for the presence of a two-state exchange provides confidence that this approach is the most robust and informative given the spectral characteristics of these proteins.

Fig. 1: Representative example of a CPMG dispersion curve showing non-zero $\Delta R_{2,eff}$, but resulting in poor fitting.

Regarding the impact of temperature, we did conduct temperature-dependent NMR experiments. While screening for an optimal temperature to perform the assignment and dynamic studies of G238S, we recorded an incomplete CPMG series at 12 °C (ν_{CPMG} points of 50 and 2000Hz) and found that the same set of residues shows a significant $\Delta R_{2,eff}$. We thus used 30 °C, which yielded the best quality spectra for a complete CPMG series and allowed performing the analysis reported in the manuscript.

EDITORIAL NOTE: REDEACTED DUE TO UNPUBLISHED MATERIAL

2. Several papers have shown that the evolutionary pathway here studied is not only dominated by an activity-stability tradeoff, but by strong epistatic interactions (see papers from Weinreich, Palzkill and Shoichet cited). There are no comments on this regard. The authors have studied different combinations of mutations that can reflect the impact of epistatic effects (for example, mutation G238S in the wt background, versus the E104K background).

Response: See comment #3, below.

3. The authors conclude that “In summary, mutations that define the evolution of TEM-1 to TEM-1* do not cause a global population shift between two unique conformations, a penicillinase and a cefotaximase. Instead, mutations independently optimize the population of different regions, including active site walls and scaffold sites, resulting in a combinatorial set of TEM-1 conformers from which a potent cefotaximase conformation is optimized along the pathway” Could this be interpreted as reflecting epistatic interactions?

Response: Yes, this is indeed the case. In the revised version, we explicitly mention the conformational epistasis between E104K and G238S and how this reflects in the two-site exchange of active-site walls.

However, the current NMR dataset lacks sufficient information, as it comprises only 6 of the 16 evolutionary intermediates investigated by Weinreich and other groups. Even for the three mutations considered here, we have analyzed only one of the three possible double mutants. Therefore, we cannot generalize how conformational epistasis and epistasis in k_{cat}/K_M observed across the landscape correlate.

Instead, we have highlighted that *mutations have unique control over the positions of the individual equilibria through narrow networks*. For G238S and E104K, the position of the equilibrium in the context of the E104K/G238S double mutant is determined by G238S for all active-site walls and scaffold elements, except for the SDN loop, where the position is determined by E104K. Similarly, for scaffold elements (e.g. helices a_1 and a_{12}), the position of the equilibrium in the A42G/E104K/G238S triple mutant is determined by A42G. Fig. 3 is now updated and in the relevant free energy diagram we have included E104K/G238S to better illustrate the epistatic effect.

For the same reason, our HDX data cannot provide insights into possible epistasis in local stability. In this respect, E104 and A42G do stabilize specific secondary structure elements in the G238S and E104K/G238S backgrounds, respectively, but epistatic interactions cannot be assessed unless HDX profiles are determined in additional genetic backgrounds, including E104K and A42G. As noted below, we do seek ways of obtaining local (residue resolved) $\Delta\Delta G$ s from which we hope to obtain information on epistasis in stability.

4. The manuscript clearly describes the dynamic effects of individual mutations, and relies in a large amount of previously published activity and MIC data. A recompilation of this information from the literature in a Supplementary table to illustrate quantitatively the impact of each mutation in activity and fitness would help the reader who is not acquainted with the details of evolution of TEM lactamases. Additionally, if feasible, including a column summarizing the overall dynamic impact of each mutation (e.g., $\Delta\Delta G$).

Response: The table with references has been included and cited in the *Introduction* section.

The authors choose MS instead of NMR for studying the impact of HXD based on the fact that MS affords measurements even for amide groups broadened due to exchange phenomena. NMR was used to interrogate the dynamics of lactamases from the CTX-M family. Did they give it a try?

Response: Yes, we did attempt to use NMR to measure HDX for the TEM variants; however, we ultimately chose to proceed with mass spectrometry (MS) for the following reasons:

- (1) *Increased Coverage:* As noted in the main text, MS allows us to obtain HDX information for a broader range of amide groups, including those that are broadened beyond detection due to conformational exchange phenomena and therefore not detectable by NMR. Thus, MS provides a more comprehensive view of the system's dynamics.
- (2) *Structural Rearrangements:* When attempted to monitor HDX by NMR, we observed significant chemical shift changes and, for multiple signals, peak doubling in the HSQC spectra upon rehydration of lyophilized samples in D₂O. This was the case for both the G238S and E104K/G238S variants. These observations strongly suggest that structural rearrangements occur during the rehydration process, which would complicate interpretation of an NMR-based HDX experiment. While TEM-1 did not show peak doubling, it did exhibit similar chemical shift changes (See Fig. 3, below).
- (3) *Complex Exchange Profile:* A subset of signals showed a rapid initial deuterium incorporation of approximately 20% in the first ¹⁵N HMQC experiment, followed by a plateau with no further exchange. This rapid, non-progressive exchange, combined with the observed structural changes, further substantiates that lyophilization and subsequent rehydration in D₂O alter the folding properties of the protein and reinforced our decision to use MS for reliable HDX measurements.

Despite of these challenges, we do seek ways to accurately obtain local $\Delta\Delta G$ s via either NMR or HDX.

Fig. 3: Comparison of the ^1H - ^{15}N HSQCs of G238S (left) and TEM-1 (right) acquired in H_2O and in D_2O after lyophilization. The spectra acquired in D_2O correspond to the first exchange time point (includes a deadtime of ~ 10 minutes) and are plotted at a lower threshold compared to the reference spectra. Multiple G238S signals show peak doubling, corresponding to two different conformations. Only small chemical shift perturbations were observed for TEM-1.

Reviewer #2: Arcia et al. present a comprehensive study on the role of protein dynamics in the activity-stability tradeoff of TEM-1 β -lactamase and its evolutionary adaptations toward cefotaxime resistance. Employing a combination of NMR, HDX, the authors systematically dissect the dynamic behavior of several TEM-1 variants, focusing on key mutations (G238S, E104K, A42G) and the combined effects of these mutations along the evolutionary trajectory. This approach allows for a detailed assessment of how primary and secondary mutations differentially modulate dynamics and local stability.

The study emphasizes the significance of local and global dynamics in shaping the functional landscape of TEM-1 and identifies the mutations propagate to modulate many sites around active site wall as well as remote scaffold sites. While the study provides a methodologically robust analysis of TEM-1 dynamics and the evolutionary tradeoff between activity and local stability, emphasizing the conceptual novelty and organizing key findings more clearly would significantly enhance its impact on the fields of protein engineering and antibiotic resistance research. It would be beneficial for the authors to address the following concerns and provide clarifications prior to publication:

1. The authors extensively analyze the dynamics of single and combined mutations along the evolutionary trajectory. The Weinreich group (Ref. 35) characterized 16 variants comprising all combinations of the four mutations and discussed pleiotropic effects and epistasis, measuring catalytic efficiency and thermostability (ΔG of folding). They concluded that cefotaxime resistance in TEM-1 β -lactamase is primarily driven by enhancements in catalytic activity rather than changes in thermostability. Furthermore, they noted that subsequent mutations primarily reinforce local stability or kinetic stability rather than global stability.

The manuscript uses the terms local stability and global thermodynamic stability interchangeably, which could be misleading. The authors should clearly define these terms and distinguish their roles in pleiotropy and evolutionary compensation. Additionally, clarifying how acquired mutations specifically compensate for deleterious effects would align the current findings with prior work and highlight the novel contributions of the study.

Response: Thank you for the comment.

(Global vs. Local stability) We agree that a clear distinction between local and global stability is needed to better deliver the importance of this work. The original version of the manuscript includes, in the *Introduction*, a section

on global stability, which remains as is. A direct comparison between global and local stability is now provided in the *Discussion* section. As in the original manuscript, in the revised version we still use ΔG and $\Delta\Delta G$ only to refer to global thermodynamic stability (measured by thermal or chemical denaturation). We have removed the term *non-native* when referring to the open allosteric state to avoid any misconception with a largely unfolded state.

(Pleiotropy and Compensation) We have now included connections between the local stability changes reported here for the $\alpha_1/\alpha_{11}/\alpha_{12}$ network and prior work on mutant or engineered TEM-1 constructs in the same region. These studies have shown that perturbations of this network result in (global) stability defects.

In the original manuscript we have discussed our data in the context of pleiotropic defects relating to misfolding, aggregation and degradation, mainly because these have been characterized in the past for TEM-1 and other β -lactamases. In the revised version, we now recognize that multiple other pleiotropic effects may operate differentially along the trajectory (*Discussion* section), including *kinetic stability*, the *in vivo environment* (crowding) and the *interaction with other cellular factors*. Still, their impact on fitness provided by TEM-1 remains unexplored, and we have not expanded our discussion further. In addition, in the revised *Discussion*, we cite the influential work by the Villa group, where, using HDX, it was shown that Zn^{2+} and naturally occurring variants modulate the dynamics of a long C-terminal segment of NDM metallo- β -lactamases, causing differential accessibility to proteolytic degradation and, therefore, distinct in-cell kinetic stability profiles.

More on the pleiotropy: We have significantly revised the presentation of pleiotropic effects caused by secondary mutations, that was heavily stability-oriented in the original manuscript. In the revised version we also highlight how pleiotropic effects related to activity aid to TEM-1 evolution (see the paragraph *Dynamic population shift of TEM-1 conformers along cefotaximase evolution* and the *Discussion* section). This is very clear for A42G. This mutation is considered a *stabilizing substitution*, as it is selected for enhancing stability. However, it is also present in resurrected β -lactamase sequences that behave as *generalists* (see work from Ozkan and Sanchez-Ruiz groups doi:10.1093/molbev/msu281 & <https://doi.org/10.1021/ja311630a>) and in CTX-Ms that are *bona fide* cefotaximases. Moreover, the Weinreich group reports a tenfold improved catalytic efficiency for the incorporation of A42G in the G238S or E104K/G238S backgrounds.

Under the “Dynamic population shift” paragraph we have included a series of our own activity assays covering the TEM-1 \rightarrow G238S \rightarrow E104K/G238S \rightarrow A42G/E104K/G238S trajectory and highlight that the concerted population shift of active site walls by A42G back to the original penicillinase state is indeed accompanied by improved catalytic efficiency. The same is true for the local effect of E104K on the two-state exchange of the SDN-loop. We conclude that the role of secondary substitutions like A42G in pleiotropy goes beyond correcting local stability defects, and directly affects catalytic efficiency. Importantly, it is also becoming apparent that both states are required for efficient catalysis. The model we propose for this TEM-1 trajectory is: G238S introduces and strongly biases the conformational ensemble toward the new function conformation, while E104K and A42G progressively restore a significant population of the penicillinase conformation. The new Fig. 3 and the revised text better convey the pleiotropic effects on activity, while our HDX data on stability.

2. Studies by Bowman et al. using thiol labeling and NMR addressed the dynamic interplay between closed and open conformations of the Ω -loop in β -lactamases. Beta-lactamases with a higher population of the open Ω -loop conformation exhibit increased catalytic efficiency against benzylpenicillin, a smaller β -lactam substrate, whereas the closed conformation is more favorable for the hydrolysis of larger substrates like cefotaxime. The findings in Bowman et al. appear to align with the current work, yet the manuscript does not explicitly connect these studies. Incorporating a discussion of how the observed dynamic shifts in the Ω -loop correlate with substrate specificity in the current work would strengthen the contextual framework and provide a more integrated perspective.

Response: Our current dataset agrees with the work from the Bowman group in that there is conformational exchange at the Ω - and 238-loop interface, at the C-terminal segment of the Ω -loop. We do cite and note the original thiol labeling work (*Nat Commun* 7, 12965, doi:10.1038/ncomms12965 (2016)) as well as the subsequent and highly relevant NMR work from the same group (*Proc Natl Acad Sci U S A* 118, doi:10.1073/pnas.2106473118 (2021)). Consistent with the MST data for a TEM-1-like conformation, i.e. a state where the Ω -loop samples predominantly a penicillinase state, also termed the “closed state”, we specifically note that the E166 signal of G238S HSQC shows a minimal perturbation, to emphasize that this is the preferred Ω -loop geometry for cefotaxime hydrolysis.

EDITORIAL NOTE: REDACTED DUE TO UNPUBLISHED MATERIAL

3. The linear chemical shift analysis presented in Figure 4, illustrating the introduction of E104K in the G238S background and A42G in the E104K/G238S background, is compelling. However, the text is somewhat ambiguous. While the authors state that mutations independently optimize distinct regions without inducing a global population shift between cefotaximase- and penicillinase-optimized conformations, the analysis simultaneously implies that such states exist. This dual framing is conceptually inconsistent. It would be beneficial to clearly define what constitutes a cefotaximase-optimized conformation versus a penicillinase-optimized conformation.

Response: We have now modified the text in the “Dynamic population shift of TEM-1 conformers along cefotaximase evolution” paragraph and throughout the manuscript to better reflect our findings and clarify that multiple penicillinase/cefotaximase states are formed by this series of mutants. They result by altered population ratios of individual active site walls undergoing two-state exchange and they are subject to epistatic effects. We were therefore cautious to use the terms like a penicillinase state.

We also find that the term “state” more accurately reflects the mechanistic details of this system. Mutations act as rheostats of the two-state exchange processes, and the new Fig. 3, describes better this continuum between penicillinase and cefotaximase states. A42G, for example, affects all active site wall conformers and alters the relative population ratio between the new-function conformation and the penicillinase conformation to improve catalytic efficiency.

Additionally, providing quantitative data, such as interatomic distances, dihedral angles, or root-mean-square fluctuations, would concretely delineate how these states differ.

Response: The NMR data presented here do not yield structural information. Still, in the revised manuscript we included that the structural rearrangement observed in the available structures, primarily those containing G238S and E104K/G238S, are found at the β_3 strand/238-loop.

EDITORIAL NOTE: REDACTED DUE TO UNPUBLISHED MATERIAL

Further, a clarification is needed on whether the observed shifts represent distinct, discrete states or a continuous ensemble of conformations.

Response: In the revised manuscript we have kept the original statement “*This suggests that each of these sites exists in an equilibrium between two states that interconvert in the fast exchange regime, with the position of a signal along the line marking the population weighted average of each state.*”. This best describes our observations and agrees with the CPMG data acquired with G238S, showing a two-state exchange. We cannot exclude the possibility that in addition to the two states observed here, other low populated states are also sampled. This is however the minimal set of conformers that can explain the dataset and the biochemistry of the system.

Statistical validation of the linearity of the chemical shift data should also be provided to ensure that the observed trend is not an artifact of selective data reporting or overfitting.

Response: We have a standard digital resolution for the two dimensions of the HSQC experiments (~4 Hz/point for ^1H and ~8 Hz/point for ^{15}N), so the position of a signal in the spectrum is very accurately determined. The biggest source of error in this kind of analysis typically arises from different samples and is related to small temperature or pH variations between runs. For TEM-1, G238S and E104K, that multiple spectra have been acquired as both biological and technical replicates during our studies, an RMSD of ~ 1 Hz is obtained for both the ^1H and ^{15}N dimensions. This is significantly lower compared to the CSPs reported in here as significant, which typically is in the order of > 10 Hz. Although not common in this type of analysis, we have now included the range of R^2 values obtained for multiple signals, which is from 0.82 to 0.98. We note that deviations from linearity are expected when nuclei are in the vicinity of a mutation site and therefore there is a direct impact on the chemical shift. We have tried not to select such residues.

4. The authors effectively utilize HDX data to assess how G238S induces local destabilization and how E104K and A42G subsequently mitigate these effects. However, the manuscript predominantly discusses changes in folding stability ($\Delta\Delta G$ values), which do not directly correlate with HDX data reflecting localized stability perturbations.

Response: The data presented in this study report differential deuterium incorporation ($\Delta\%$ uptake), which is certainly not local $\Delta\Delta G$, but directly reflects on changes in differential local stability. As noted above, here we have used ΔG and $\Delta\Delta G$ only when referring to global stability.

In general, HDX requires an unfolding event that breaks H-bonding and allows exposure to solvent. Hence, rates depend on the equilibrium between folded and locally unfolded states. While the set of our HDX data does not allow extraction of residue level ΔG values, we still can use our peptic peptide level resolution to assess local stability. This is further supported by the available structures of the evolutionary intermediates studied in here , where the structural rearrangements observed cannot account for enhanced deuterium incorporation due to changes in secondary structure or major structural rearrangements, but truly reflect changes in local stability and dynamics. We now explicitly mention in the manuscript that *$\Delta D\%$ uptake is used as a proxy of local stability (see paragraph Local stability changes reveal unique G238S-induced pleiotropic defects).*

The manuscript would be strengthened by integrating additional analyses that specifically quantify changes in local stability, potentially including entropy-enthalpy compensation to capture how different regions are variably affected by each mutation.

Response: The current set of HDX data does not allow accurate determination of local ΔG values at the residue level. This would require a significantly larger pool of fragments than is currently available using only pepsin. We will agree that enthalpy-entropy compensation will be apparent in the differential stability distribution between mutants -as is in many other biophysical measurements- but assessing it would require HDX datasets acquired at different temperatures, and this is beyond the purpose of this work.

Additionally, molecular dynamics simulations, as reported in Yu et al. (Proc. Natl. Acad. Sci. USA 2018) and Ose et al. (Biophys. J. 2023), provide supporting evidence for the findings in Figures 5 and 6. Citing these studies and discussing how they align with the current work would provide a more comprehensive contextual framework.

Response: These are indeed very relevant papers and are now cited in both the *Introduction* and the *Discussion*.

5. The identification of a cryptic allosteric site formed by the $\alpha 11$ - $\alpha 12$ interface is briefly mentioned but not sufficiently analyzed mechanistically. The cited works also do not provide a clear mechanistic explanation. How this site influences catalytic efficiency and inhibition remains unclear. Is the formation of this cryptic site a compensatory mechanism or an unintended structural consequence of the mutations? More direct evidence is needed to substantiate the proposed role of this site, potentially

including data on how ligand binding, mutational analysis, or dynamic perturbations influence its functional role.

Response: The cited papers are on the work demonstrating that the TEM-1 allosteric site pre-exists as a sparsely populated excited state. It also includes the structural work from the Shoichet lab, where the different modes of binding for core disruptive ligands were identified. The HDX data presented in here indicate that G238S destabilizes helices α_1 , α_{11} and α_{12} , as well as strands β_3 and β_4 (Fig. 4 and Supplementary Fig. 5). Therefore, we hypothesized that this would lead to partial shift of the allosteric site from the closed to the open state.

In the revised manuscript we include NMR and kinetic data acquired with FTA, one of the core disruptive ligands (allosteric inhibitors), previously identified (Fig. 5). Consistent with our hypothesis, we find that FTA causes minimal perturbations in the NMR spectrum of TEM-1 and exhibits weak inhibition against nitrocefin hydrolysis indicating that the equilibrium is strongly biased towards the closed state. G238S, however, shows global and large changes in chemical shifts that localize at the predicted opening of the fold, suggesting that the mutation acts by partially destabilizing the closed state, and therefore is significantly more susceptible to FTA inhibition. Still, the strong stabilizing effect of E104K and A42G on helices α_{12} and α_1 is not enough to reverse this (apparently small) population shift. We speculate that this is because strands β_3/β_4 and helix α_{11} are not stabilized by the secondary substitutions. Therefore, the double and triple mutant show lower albeit comparable IC_{50} to that of G238S (Supplementary Fig. 5b). These data are now incorporated in the revised version. Certainly, more effort (structural studies with TEM-1 mutants) is required to fully understand its function and exploit it pharmacologically. We did not manage to achieve this during the revision cycle.

6. The authors assert that their findings offer actionable strategies for enhancing enzyme function while mitigating stability tradeoffs. However, the broader applicability of this framework beyond TEM-1 is not clearly articulated. Explicitly discussing how the dynamic tradeoff framework developed here could be applied to other enzymes, particularly those with known allosteric networks or cryptic binding sites, would broaden the impact of the study and reinforce its relevance to protein engineering and evolutionary biology.

Response: As noted, the general impact in (protein) evolution biology, is that the use of solution techniques that provide adequate resolution can inform on the molecular basis of the activity-stability tradeoff in other systems. We expect that the dynamic properties and extent of population shifts between states, as observed here for TEM-1, to differ significantly from system to system. However, we believe that by integrating tools to identify the scaffold residue networks that are subject to stability defects, such as HDX, can guide the thermostability problem of engineered or designed proteins.

Reviewer #3: Review for Arcia et al., “Dynamic signature of activity-stability tradeoff in lactamase evolution”

Arcia et al. describe how mutations in the early evolution of antibiotic resistance in TEM-1 beta lactamase influence the protein’s substrate specificity and overall conformational ensemble properties. They use a combination of NMR spectroscopy and HDX/MS to achieve high spatiotemporal resolution in their study of the allosteric effects in the protein ensemble.

It’s a very interesting paper, but I focus here only on the HDX/MS data analysis. There are some errors that affect the interpretation of the data. Some of them might come from errors in offsetting the residue indexing in the analysis. One would assume that the authors meant to maintain indexing that places G238S at position 238, but the indexing is inconsistent no matter what the reference is. Apart from the ambiguity in the annotation, in several cases the authors actually have the wrong number of residues in a peptide, which means that their reported %D is very likely incorrect, if the data themselves are correct. Additionally, this means that some of the peptides are not possible to correctly identify, because if the residue numbering is correct, then the data would correspond to the reported peptide, but if the residue numbering is off by -3 to +3 as in some of the other peptides, then the reported data correspond to a different peptide altogether.

Response: We first note that the constructs used in this study produce periplasmic TEM proteins with native N-terminus starting at position H26. Throughout this work residue numbering is according to the Ambler scheme for class A β -lactamases (PMID: 2039479, Ambler et al., Biochem. J. 276, 269-272, 1991), which places the active site serine at number 70. For TEM-1, this numbering includes gaps between residues G238-E240 and P252-D254, i.e. there are no residues 239 and 253. This important information is now provided early in the manuscript (first paragraph), as well as under the HDX methods paragraph.

We have now checked all the peptides reported in the manuscript and identified two peptides presented in Fig. 5 and Fig. 6 (31-40 and 214-221), and one presented in the Supplementary Fig. 5 (31-40), for which the numbering given is not correct, although the sequence is and it was correctly identified (see below in detail).

In addition, (a) we have corrected the sequence of peptide #45 in Supplementary Fig. 5 (a T was missing at the C-terminus of the sequence) and (b) corrected the legend for Supplementary Fig. 5b, as G238S is shown in blue.

In Fig. 5b:

- Peptide 32-40 should be 31-40: VKVKDAEDQL corrected
- Peptide 163-182: DRWEPELNEAIPNDERDTTM is correct no change
- Peptide 211-221 should be 214-221: DKVAGPLL corrected
- Peptide 222-230: RSALPAGW is correct no change
- Peptide 230-248 is ambiguous: it is possible that it should be 230-247: FIADKSGAGERGSRGIIA given the indexing problem, or alternatively, it might be correctly identified as 230-248: FIADKSGASERGSRGIIAA (it’s not clear which peptide these data correspond to) 230-248 is correct (one gap is included)
- Peptide 263-269 should be 261-267: IYTTGSQ 263-269 is correct (two gaps are included)

- Peptide 273-283 should be 271-281: DERNRQIAEIG 273-283 is correct (two gaps are included)
- Peptide 286-290 should be 284-288: LIKHW 286-290 is correct (two gaps are included)

Also in Fig. 5c, the authors might want to add the yellow sphere mentioned in the caption to the figure. The opacity of the surface representation is now adjusted to make the position of the mutation site apparent.

There are also mistakes in Fig. 6b:

- Peptide 32-40 should be 31-40 corrected
- Peptide 163-182 is correct no change
- Peptide 211-221 should be 214-221 corrected
- Peptide 222-230 is correct no change
- Peptide 230-248 is ambiguous: if the peptide sequence is identified correctly, then it should be 230-247, but if the indexing is correct, it should likely correspond to peptide 230-248: FIADKSGAGERGSRGIIAA 230-248 is correct (one gap included)

also, it is not proper to compare the deuterium uptake for two peptides where there is a difference in the sequence on the same graph, because the k_{int} for the mutated residue will be different and we cannot account for the magnitude of this difference in the %D – the authors should really find another way to show the data for these peptides

Yes, the k_{int} of the mutated residue will differ. In fact, the mutation will also affect the k_{int} of the adjacent residue on its C-terminal side. According to Englander's predictive formulas, the k_{int} of the mutated residue is expected to increase by approximately 2.5x, while that of its C-terminal neighbor may increase by about 1.5x. As a result, we agree that the variations in k_{int} could introduce a measurable change in deuterium uptake that is unrelated to changes in local structural stability or dynamics.

However, this change in deuterium uptake should not exceed ~1.5 Da, even under the most unfavorable (yet unlikely) scenario in which the backbone amide hydrogens of both affected residues are fully solvent-exposed and not engaged in hydrogen bonding (assuming an average back-exchange of 25%, too). Still, we observed that the mutation leads to an increase of ~ 3.3 Da in deuterium uptake after 1000 seconds of exchange and never less than 2.5 Da across all measured HDX time points. Therefore, we believe that it remains appropriate to include the $\Delta D\%$ for these two or other peptides corresponding to this region, given the magnitude of the observed differences in deuterium uptake among these evolutionary intermediates.

- Peptide 263-269 should be 261-267 263-269 is correct (two gaps included)
- Peptide 273-283 should be 271-281 273-283 is correct (two gaps included)
- Peptide 286-290 should be 284-288. 286-290 is correct (two gaps included)

Did not look at the supplement but the authors should double-check all their peptides. As noted above, the supplementary information is also corrected.

Reviewer #1: The authors have adequately addressed all my concerns, providing new experiments, and appropriate explaining the experimental limitations to perform some assays. Overall, this resulted in a great manuscript that will be impactful in the field.

Reviewer #2: The authors addressed all the questions I raised.

Reviewer #3: Thank you to the authors for correcting the mistakes in their uptake plot annotations. But the authors still did not correct the uptake plots showing peptides of different sequences together on the same axes. This is incredibly improper, and there are a lot of problems with their handwaving explanation about why this is OK - it is not OK, and it sets a bad precedent in the literature for our field. The biggest problems with their explanation: 1) mutations affect both N- and C-terminal residues' exchange rates (see recent work from Hamuro showing this in a tabular format, JASMS); 2) even single mutations often change the local/global structure of the protein, which commonly manifest as changes in forward AND back exchange. The authors do not take back-exchange into account in this work. So it would be great if the authors could please treat the data properly and put the uptake for peptides of different sequences in different plots. Thanks.

Response: The following edits were made:

We have removed the panel showing the comparison of deuterium uptake for the peptide that contains the G238S mutation and an adjacent peptide was included instead. We now compare the uptake of the wild-type and mutant peptide of the G238-loop in different plots (side-by-side) in a new panel of Supplementary Fig. 4d. Similarly, the peptide with another mutation site (E104K) was removed from Supplementary Fig. 6b.